EMBO
Molecular Medicine

# Alternative oxidase-mediated respiration prevents lethal mitochondrial cardiomyopathy

Jayasimman Rajendran[1,2], Janne Purhonen[1,2], Saara Tegelberg[1,3,4], Olli-Pekka Smolander[5], Matthias Mörgelin[6], Jan Rozman[7,8], Valerie Gailus-Durner[7], Helmut Fuchs[7], Martin Hrabe de Angelis[7,8,9], Petri Auvinen[5], Eero Mervaala[10], Howard T Jacobs[5,11], Marten Szibor[5,11], Vineta Fellman[1,3,12] & Jukka Kallijärvi[1,2,*] (iD)

## Abstract

Alternative oxidase (AOX) is a non-mammalian enzyme that can bypass blockade of the complex III-IV segment of the respiratory chain (RC). We crossed a *Ciona intestinalis* AOX transgene into RC complex III (cIII)-deficient *Bcs1l*[p.S78G] knock-in mice, displaying multiple visceral manifestations and premature death. The homozygotes expressing AOX were viable, and their median survival was extended from 210 to 590 days due to permanent prevention of lethal cardiomyopathy. AOX also prevented renal tubular atrophy and cerebral astrogliosis, but not liver disease, growth restriction, or lipodystrophy, suggesting distinct tissue-specific pathogenetic mechanisms. Assessment of reactive oxygen species (ROS) production and damage suggested that ROS were not instrumental in the rescue. Cardiac mitochondrial ultrastructure, mitochondrial respiration, and pathological transcriptome and metabolome alterations were essentially normalized by AOX, showing that the restored electron flow upstream of cIII was sufficient to prevent cardiac energetic crisis and detrimental decompensation. These findings demonstrate the value of AOX, both as a mechanistic tool and a potential therapeutic strategy, for cIII deficiencies.

**Keywords** BCS1L; complex III; GRACILE syndrome; mitochondrial disorder; respiratory chain
**Subject Categories** Cardiovascular System; Genetics, Gene Therapy & Genetic Disease

See also: **A Saada** (January 2019)

## Introduction

Mitochondrial disorders are the most common class of inherited errors of metabolism. However, effective treatments are lacking, and their clinical management remains largely supportive (Pfeffer *et al*, 2013). In patients with RC cIII (ubiquinol:cytochrome *c* oxidoreductase) deficiency, mutations in several genes encoding either cIII subunits or assembly factors have been identified. These compromise cIII enzymatic activity and result in a wide variety of clinical manifestations (Fernandez-Vizarra & Zeviani, 2015). *BCS1L* mutations are the most common cause of cIII deficiency, with various neonatal and adult phenotypes described worldwide (Fernandez-Vizarra & Zeviani, 2015), the most severe and prevalent of them being GRACILE syndrome (fetal growth restriction, aminoaciduria, cholestasis, liver iron overload, lactic acidosis, and early death during infancy) (Fellman *et al*, 1998; Visapää *et al*, 2002). BCS1L is a mitochondrial inner membrane translocase required for Rieske iron–sulfur protein (RISP, UQCRFS1) topogenesis and incorporation into cIII (Nobrega *et al*, 1992; Cruciat *et al*, 1999). Homozygous *Bcs1l*[c.A232G] (*Bcs1l*[p.S78G]) knock-in mice bearing the GRACILE syndrome-analogous mutation recapitulate many of the clinical manifestations, including growth failure, progressive hepatopathy, kidney tubulopathy, and, in a C57BL/6JCrlBomTac background, short survival of 35 days (Levéen *et al*, 2011; Kotarsky *et al*, 2012;

1 Folkhälsan Research Center, Helsinki, Finland
2 Clinicum, Faculty of Medicine, University of Helsinki, Helsinki, Finland
3 Department of Clinical Sciences, Lund, Pediatrics, Lund University, Lund, Sweden
4 Molecular Neurology Research Program and Neuroscience Center, University of Helsinki, Helsinki, Finland
5 Institute of Biotechnology, University of Helsinki, Helsinki, Finland
6 Division of Infection Medicine, Clinical Sciences, Lund University, Lund, Sweden
7 German Mouse Clinic, Institute of Experimental Genetics, Helmholtz Zentrum München, German Research Center for Environmental Health, Neuherberg, Germany
8 German Center for Diabetes Research (DZD), Neuherberg, Germany
9 Chair of Experimental Genetics, Center of Life and Food Sciences Weihenstephan, TU Munich, Freising-Weihenstephan, Germany
10 Department of Pharmacology, Faculty of Medicine, University of Helsinki, Helsinki, Finland
11 Faculty of Medicine and Life Sciences, University of Tampere, Tampere, Finland
12 Children's Hospital, Helsinki University Hospital, University of Helsinki, Helsinki, Finland
*Corresponding author. Tel: +358 504487006; E-mail: jukka.kallijarvi@helsinki.fi

Rajendran *et al*, 2016; Purhonen *et al*, 2017). In the slightly different C57BL/6JCrl substrain, the homozygotes develop the same early manifestations but do not succumb to the early metabolic crisis. This extends their survival to over 150 days (Purhonen *et al*, 2017) and brings additional later-onset phenotypes, such as cerebral astrogliosis (Tegelberg *et al*, 2017).

Under physiological conditions, quinols that transport electrons in the mitochondrial inner membrane are efficiently oxidized by cIII, with electron transfer via cytochrome c and cytochrome c oxidase (complex IV, cIV) to oxygen (Brand, 2010; El-Khoury *et al*, 2014). However, plants and some lower organisms, but not mammals, express alternative oxidases (AOXs) that transfer electrons directly from quinols to oxygen without proton translocation. Their main role is to maintain electron flow when the cIII-cIV segment of the RC is impaired, limiting production of ROS and supporting redox and metabolic homeostasis (McDonald & Vanlerberghe, 2004; El-Khoury *et al*, 2014). *Ciona intestinalis AOX* has been cloned and expressed in human cultured cells (Hakkaart *et al*, 2006), fruit flies and mice (El-Khoury *et al*, 2013; Szibor *et al*, 2017). In these models, AOX is inert under non-stressed conditions, most likely because it accepts electrons only when the quinone pool is highly reduced (Hoefnagel & Wiskich, 1998; Castro-Guerrero *et al*, 2004), such as under inhibition or overload of cIII or cIV (Dassa *et al*, 2009). Accordingly, upon inhibition of cIII or cIV by mutations or chemical inhibitors, ectopic AOX can maintain respiration and prevent cell death (Dassa *et al*, 2009; Fernandez-Ayala *et al*, 2009). We set out to test whether AOX expression could prevent the detrimental effects of cIII deficiency in a mammalian model, by restoring electron flow upstream of cIII. To this end, we crossed mice carrying a broadly expressed *AOX* transgene (Szibor *et al*, 2017) with the *Bcs1l*$^{c.A232G}$ mice and assessed disease progression, organ manifestations, and metabolism in the homozygotes with and without AOX expression.

# Results

### Broadly expressed AOX triples the life span of cIII-deficient *Bcs1l*$^{p.S78G}$ mice

To assess the effect of cIII bypass on the survival and tissue manifestations in cIII-deficient mice, we bred cohorts of wild-type and *Bcs1l* mutant mice with or without a *Ciona intestinalis* AOX transgene. Hereafter, we will refer to the *Bcs1l*$^{p.S78G}$ homozygotes as GRAC (as an abbreviation of GRACILE syndrome) mice. The *Bcs1l*$^{p.S78G}$ homozygotes carrying AOX transgene will be referred to as GROX mice (GRAC + AOX). Figure 1A shows a timeline of the appearance of the previously reported and novel phenotypes in GRAC mice, as well as the assessments included in this study. The GRAC mice reached the criteria of euthanasia between postnatal day 180 (P180) and P220, with median survival to P210 (Fig 1B). In contrast, the GROX mice showed no signs of terminal deterioration or spontaneous deaths at P200 and survived to a median age of 590 days (Fig 1B). To assess whether the extended survival was due to an overall improvement in energy metabolism, we measured growth, whole-body metabolism, and body composition in young adult mice. The GRAC mice were growth restricted (Fig 1C and E) and had increased lactate-to-glucose ratio (Fig 1D), low fat mass

(Fig 1F), bone density (Fig 1G), and heat production (Fig 1H) and, in females, low respiratory exchange ratio (Fig 1I). Unexpectedly, AOX had no or only small effect on these parameters (Fig 1C–I), suggesting that the AOX-mediated extension of survival depended on a tissue or cell-type specific pathology rather than whole-body energy metabolism.

### AOX permanently prevents lethal cardiomyopathy and alleviates renal and cerebral manifestations

Histopathological analysis of autopsy samples from end stage (P200) mice showed two novel phenotypes not previously reported in studies of younger GRAC mice: cortical kidney atrophy and cardiomegaly with dilated ventricles (Fig 2A and B). Fibrosis was prominent in liver, kidney, and heart (Fig 2C and D). Suspecting cardiomyopathy as the cause of death, we assessed cardiac functions at several time points. Echocardiography showed minimal functional changes at P150 (Fig EV1A–F), but severe dilated cardiomyopathy, focal fibrosis, decreased ejection fraction, and fractional shortening at P200 (Fig 2A and F), indicating end-stage cardiomyopathy. Strikingly, the GROX littermates had normal heart size (Fig 2A and B), no fibrosis (Fig 2C and D), and overtly normal cardiac function (Fig 2E and F), explaining their extended survival. mRNA expression of key markers for cardiac hypertrophy and fibrosis was significantly altered already in the presymptomatic (P150) GRAC hearts, and these changes were largely prevented in the GROX mice (Fig 2G). A kidney stress test with salt-enriched (6% w/w) chow starting at P150 had no effect on cardiac function at P200, or on survival, and blood pressure was only increased at the end stage in the salt-fed mice when compared to P150 baseline (Fig EV1A–F), consistent with a primary cardiomyopathy. Remarkably, the GROX mice had normal-sized and non-fibrotic heart throughout their life span, to over P600 (Fig EV2A–D).

Kidneys of GRAC mice showed proximal tubulopathy with fibrosis (Fig 2C and D) and tubular degeneration (decreased tubular mass) at P200 (Appendix Fig S1). In GROX mice, the kidney mass and apparent tubular volume were preserved (Fig 2A and B). On the basis of the proliferation marker Ki67 and apoptosis marker cleaved caspase-3, this was likely due to decreased apoptosis rather than increased regeneration (Appendix Fig S1I and J). However, AOX had only minor effect on other histological lesions (Appendix Fig S1A–G) or functional parameters: albuminuria, hematuria, and urinary creatinine (Fig 2H and I, Appendix Fig S1H). At P600, the kidneys were severely fibrotic but still of normal size, indicating long-term protection from tubular atrophy (Fig EV2A, B and D). Surprisingly, AOX had no effect on the liver fibrosis (Fig 2C and D) or the elevated liver enzymes (Fig 2J and K) at end stage. The cause of death of the GROX mice remains unknown, but eventual deterioration due to the progressing kidney and liver disease is an obvious possible explanation.

GRACILE syndrome patients have no encephalopathy, but the brains of GRAC mice show peculiar focal astrogliosis in the primary barrel field of the somatosensory cortex (S1BF) (Tegelberg *et al*, 2017). Staining for glial fibrillary acidic protein (GFAP) showed that the astrogliosis was almost fully prevented by AOX at P200 (Fig 3A).

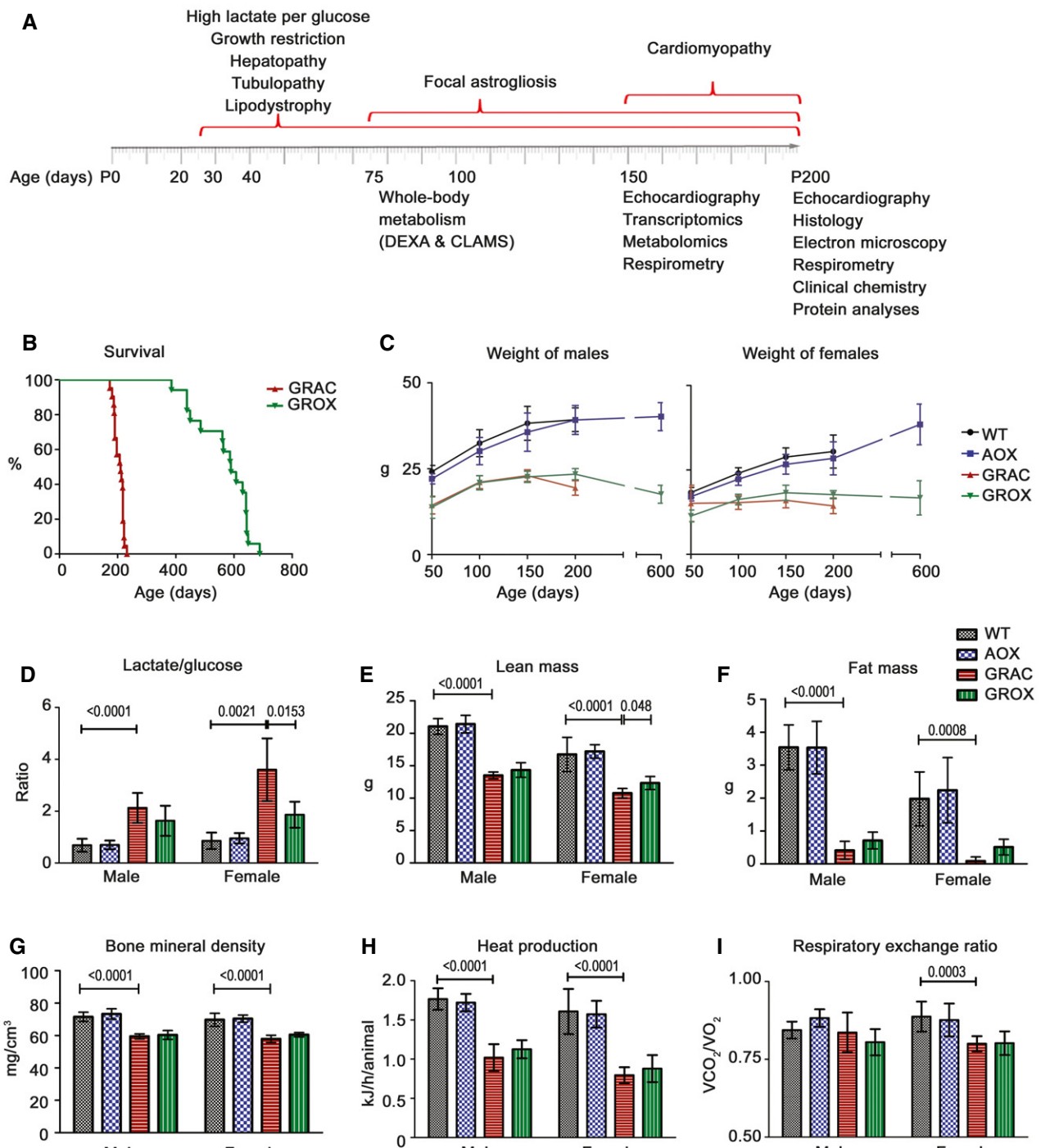

**Figure 1. AOX expression prolongs the survival of cIII-deficient *Bcs1l^{p.S78G}* mice without affecting growth or whole-body metabolism.**

A    Schematic presentation of the multiorgan manifestations, described in this study or previously, in homozygous *Bcs1l^{c.A232G}* (GRAC) mice, and the time points of the investigations performed in this study.

B    Survival curves of homozygous *Bcs1l* mutant mice without (GRAC) and with (GROX) alternative oxidase (AOX) expression (*n* = 18–21/group). The median survival of GRAC mice was 210 days and of GROX mice was 589 days with no gender difference.

C    Weight of mice from P50 to P200 (*n* > 10/group) and at P600 (*n* = 4–6/group).

D    Blood lactate-to-glucose ratio at P200 (*n* = 8/group).

E–G    Dual-energy X-ray absorptiometry (DEXA) analysis (*n* = 10/group) of (E) lean mass, (F) fat mass, and (G) bone mineral density at P98.

H, I    Indirect calorimetric measurement (*n* = 10/group) of (H) heat production and (I) respiratory exchange ratio at P77.

Data information: The survival data were analyzed using log-rank (Mantel–Cox) test (*P* < 0.0001). Bar graphs represent mean ± SD. The data (D–I) were analyzed using Kruskal–Wallis and Mann–Whitney *U*-tests for selected comparison. Significant differences between groups (*P*-value) are indicated on graphs.

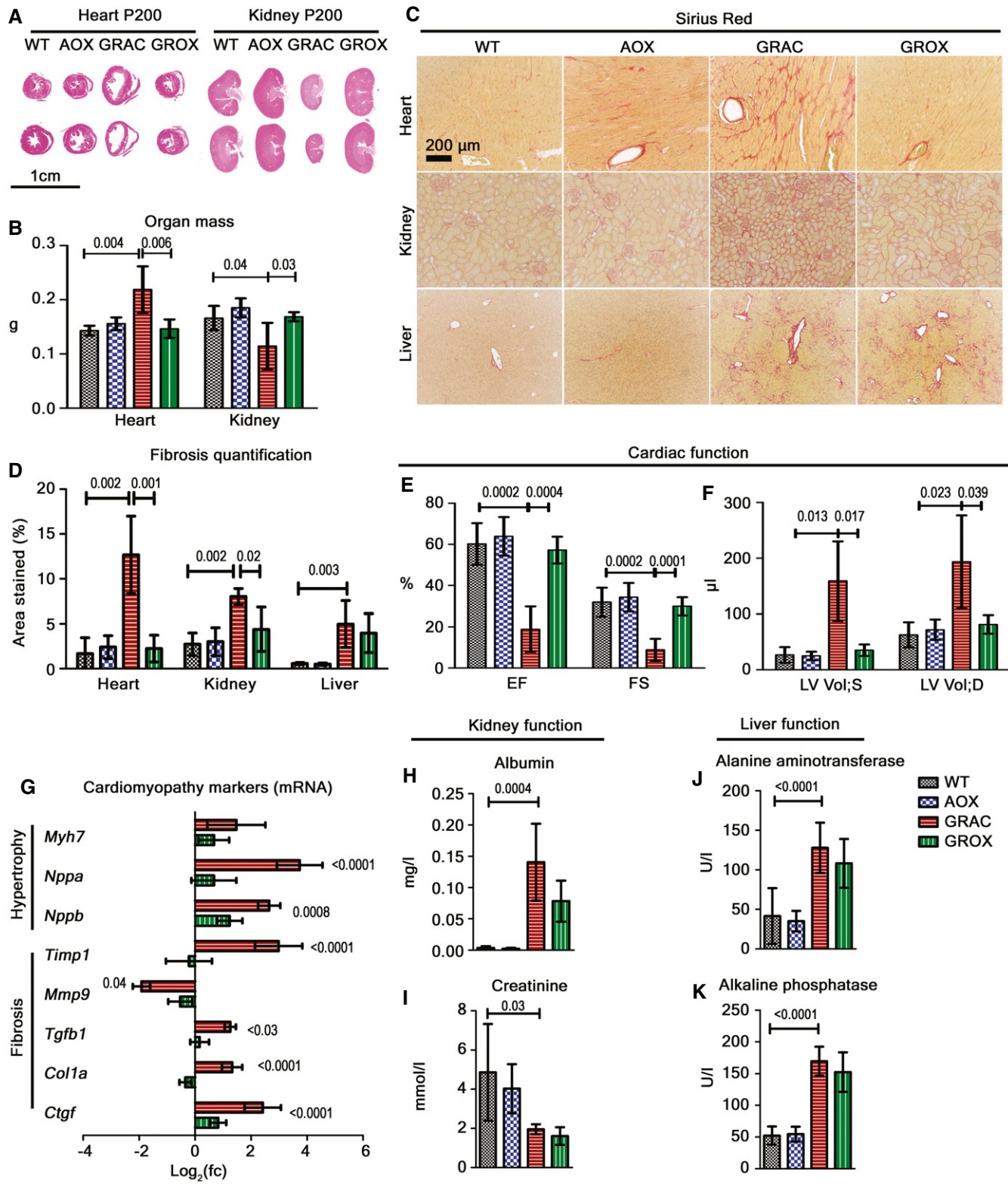

Figure 2.

## AOX preserves mitochondrial ultrastructure in rescued tissues

Previous studies have shown disrupted hepatic mitochondrial ultrastructure in younger *Bcs1l* mutant mice (Levéen *et al*, 2011;

Purhonen *et al*, 2017). In the P150 GRAC mice, electron microscopy showed abnormal mitochondrial ultrastructure in hepatocytes as well as in kidney tubular cells, and, importantly, in cardiomyocytes already at onset of the cardiomyopathy (Fig 3B). The mitochondria

**Figure 2. AOX prevents lethal cardiomyopathy and progression of renal tubulopathy to kidney atrophy.**

A    Hematoxylin–eosin-stained cross sections of heart and kidney at P200.
B    Weight of heart and kidney at P200 ($n$ = 4/group).
C, D    Sirius Red staining (C) for fibrosis in liver, heart, and kidney at P200, and (D) quantification of fibrosis in myocardium, kidney cortex, and liver ($n$ = 5–7).
E, F    Echocardiography data ($n$ = 4–6/group) showing (E) ejection fraction (EF) and fractional shortening (FS). (F) Systolic (LV Vol;S) and diastolic (LV Vol;D) left ventricle volume in mice.
G    Expression of cardiac hypertrophy and fibrosis-associated genes in the presymptomatic (P150) heart ($n$ = 6/group).
H, I    24-h excretion of (H) albumin and (I) creatinine in urine at P200 ($n$ = 4/group).
J, K    Liver enzymes (J) alanine aminotransferase (ALT) and (K) alkaline phosphatase (ALP) in plasma at P200 ($n$ = 8/group).

Data information: Bar graphs represent mean ± SD. Statistics: one-way ANOVA followed by Tukey's test (for graphs B, D, G, H, J, and K), one-way ANOVA followed by unpaired $t$-test with Welch's correction (for graphs E and F), and Mann–Whitney $U$-tests (for graph I).

were smaller and contained fewer cristae, which were thicker than in WT mitochondria. AOX fully or partially prevented these changes in cardiomyocytes and kidney tubular cells, but not in hepatocytes, (Fig 3C–F), correlating faithfully with the rescue of tissue pathology.

### AOX relieves cardiac metabolic stress and prevents decompensation

Seeking a mechanism for the remarkably tissue-specific rescue effects of AOX, we performed transcriptomics and metabolomics at P150, at the onset of the cardiomyopathy. The three affected tissues showed marked global transcriptional changes, including in the GRAC heart despite it being functionally normal at this stage (Fig 4A–C). In correlation with the histological findings, AOX normalized the expression of only few genes in the liver, but of about 25% of dysregulated genes in the kidney and about 50% in the heart (Appendix Fig S2A–C). The most robustly upregulated gene sets were related to extracellular matrix organization, a signature of tissue remodeling and fibrosis (Fig 4D–F, Appendix Table S1), and cell cycle in liver and kidney, but not in the heart (Fig 4D–F) reflecting differential regenerative capacity of these tissues. As expected, energy metabolism-related gene expression was altered in all three tissues (Fig 4D–F, Appendix Fig S2D). Alternative oxidase almost fully prevented these changes in heart, but not in kidney or liver (Fig 4D–F). Notably, AOX had a significant effect on gene expression in the Bcs1l wild-type heart, including upregulation of genes related to mitochondrial function (Fig 4D, G, I and Appendix Fig S2D). Expression of the major cardiac metabolic regulator HIF-1α and the metabolic stress-inducible transcriptional regulators ATF3 and ATF4 (Kalfon et al, 2017; Quiros et al, 2017) was elevated in GRAC hearts and normalized by AOX (Fig 4J and K), indicating relieved metabolic stress. Upregulation of PGC-1α, the master regulator of mitochondrial biogenesis, and the mitochondrial transcription factor TFAM, in the GRAC heart suggested an attempt for compensatory mitochondrial biogenesis (Fig 4G–I). However, Western blot analysis using the abundance of RC complex subunits as a proxy showed no significant changes in mitochondrial mass between the groups (Fig EV3I) in any tissue. We previously identified an upregulated set of genes, which we designated cIII stress signature, in P45 Bcs1l mutant livers (Purhonen et al, 2017). This gene set was also highly upregulated in all three GRAC tissues. Despite rescue of kidney tubular mass, AOX amplified the cIII stress signature specifically in this tissue, but not in the heart or liver (Fig 4L).

Alternative oxidase is under the strong synthetic CAG promoter and expressed in all tissues in the $Rosa26^{AOX}$ mice (Szibor et al, 2017). In our mice, AOX mRNA expression was similar in heart and liver and somewhat lower in kidney (Fig EV3H). In total tissue lysates from the AOX mice, the amount of AOX protein was considerably higher in heart than in liver or kidney (Fig EV3H). However, this difference was mainly due to the higher mitochondrial mass in heart, as shown by the mitochondrial loading control VDAC1 and also by most respiratory chain subunits (Fig EV3I). Interestingly, the amount of AOX protein was affected by the Bcs1l mutation (AOX vs. GROX mice) so that the amount in the three GROX tissues was almost identical (Fig EV3H), which essentially rules out that the differences in rescue would be due to different levels of AOX expression.

The metabolomics revealed only modest cardiac metabolite changes at the onset (P150) of the cardiomyopathy (Fig 5A, Appendix Table S2). Nevertheless, several three-carbon glycolytic intermediates were depleted, and these tended to be normalized by AOX (Fig 5A). In line with this, the gene expression of PPAR-1α, the major driver of fatty acid utilization, was downregulated in GRAC but normal in GROX heart (Fig 4G). The TCA cycle metabolites malate and fumarate, adenylate energy charge, and NADH/NAD$^+$ ratio, which all could be affected by both the cIII blockade and AOX, were not changed in the GRAC or GROX heart tissue (Fig 5C–F). Interestingly, the amino acid proline, which has been long known to accumulate in conditions with lactic academia (Kowaloff et al, 1977), was below detection limit in WT heart but increased to approximately 200 nmol/g in GRAC heart. The proline accumulation was partially prevented by AOX (Fig 5G). Concurrently, glutamate, the biosynthetic precursor of proline, was decreased to about 50% in GRAC mice and normalized to WT level in GROX heart tissue (Fig 5A). The mRNA levels of glutamate γ-semialdehyde synthetase (P5CS, ALDH18A1) and pyrroline-5-carboxylate reductase 1 (P5CR, PYCR1), both driving proline synthesis from glutamate, were upregulated 1.8-fold and 1.7-fold, respectively, in the GRAC heart (Fig 5H and I).

Targeted metabolomics of liver tissue (Fig 5B, Appendix Table S2) showed significant metabolite changes characteristic of failing energy metabolism and glycogen depletion, as previously shown in juvenile mice (Kotarsky et al, 2012). These included decreased hexose phosphates, several other glycolytic intermediates, acetyl-CoA and NAD$^+$, and elevated TCA cycle intermediates and amino acids, the latter suggesting protein degradation for fuel. NADH/NAD$^+$ ratio was increased (Fig 5F), in line with our previously published data showing decreased hepatic NAD$^+$ (Purhonen et al, 2018). Proline level was not changed in liver (Appendix Table S2). Alternative oxidase had only a minor effect on the hepatic metabolite levels (Fig 5B, E, and F).

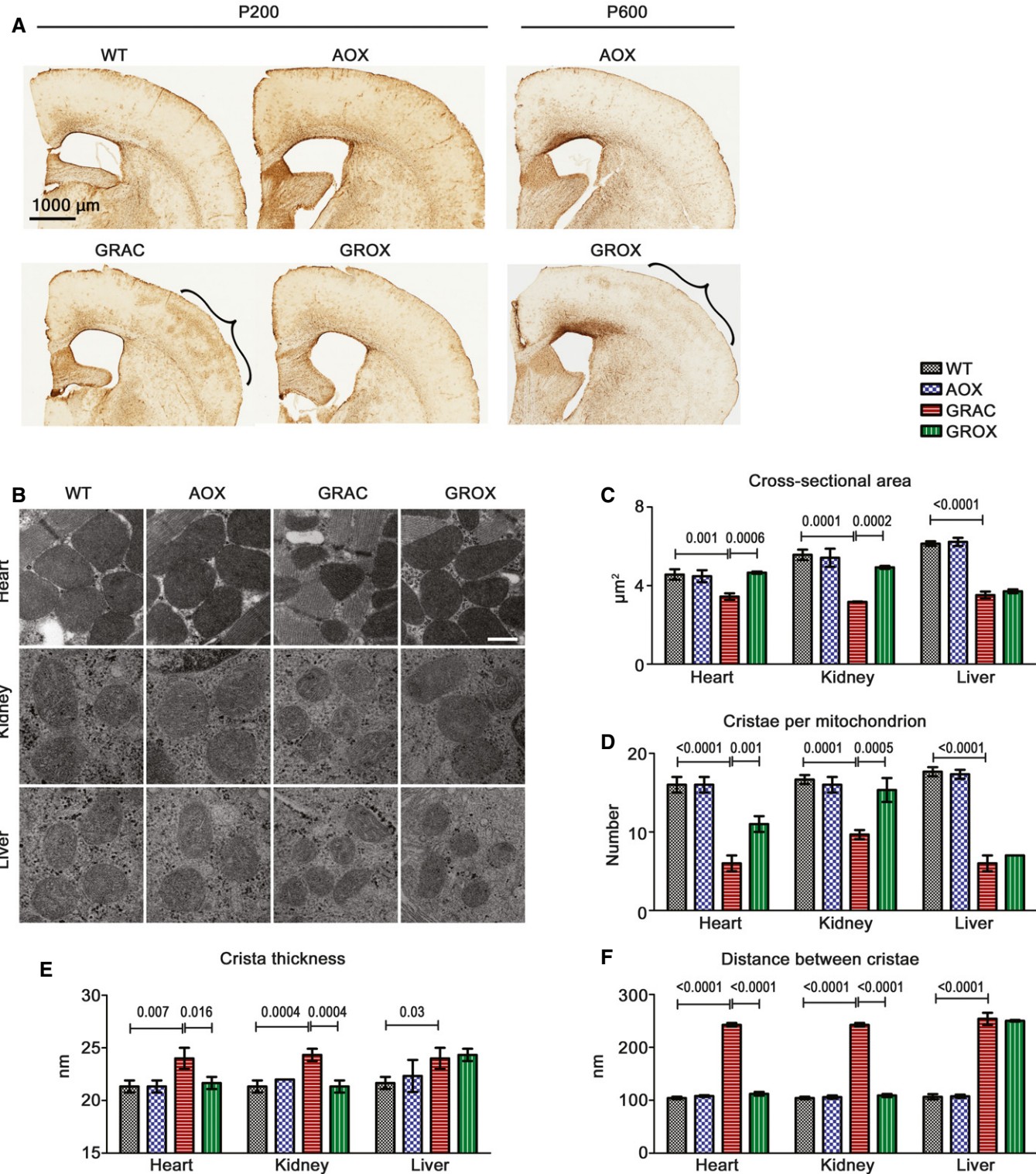

**Figure 3. AOX ameliorates cerebral astrogliosis and maintains mitochondrial ultrastructure in rescued tissues.**

A  GFAP staining for cerebral astrocytes. The barrel field of the primary somatosensory cortex (S1BF) is highlighted with brackets.

B  Mitochondrial ultrastructure in cardiomyocytes, kidney tubular cells, and hepatocytes at P200 as visualized by electron microscopy. Scale bar 1 μm.

C–F  (C) Average cross-sectional area of mitochondrion, (D) number of cristae per mitochondrion, (E) crista thickness, and (F) average distance between cristae in mitochondria (n = 3 mice/group).

Data information: Bar graphs represent mean ± SD. Statistics for graphs (C–F): one-way ANOVA followed by Tukey's test.

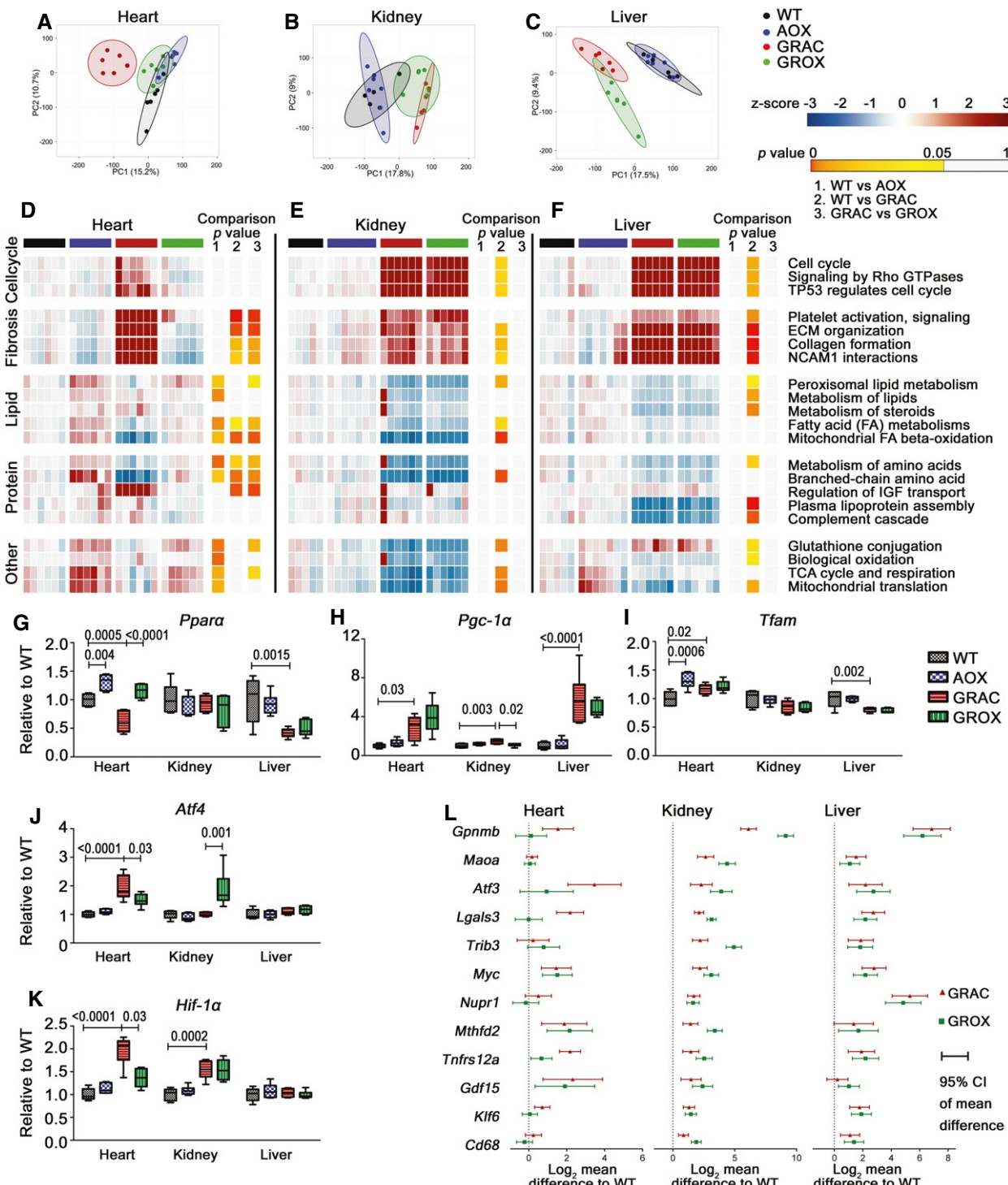

**Figure 4.  AOX mitigates cardiac, but not hepatic or renal, metabolic stress-related gene expression changes.**

A–C   Principal component analysis of transcriptome data from (A) heart, (B) kidney, and (C) liver.

D–F   Heat map visualization of pathway enrichment analysis (Reactome database). Full pathway analysis is provided in Appendix Table S1. Benjamin–Hochberg FDR-corrected *P*-values are color labeled as indicated for the three comparisons.

G–K   Gene expression of major transcriptional regulators of energy metabolism (PPAR-α, HIF-1α), mitochondrial biogenesis (PGC-1α, TFAM), and stress responses (ATF4) in heart at P150 (*n* = 6/group).

L       Expression of cIII stress signature genes in heart, kidney, and liver (*n* = 6/group). Error bars represent 95% confidence interval of mean difference.

Data information: Box plots (G–K) represent quartiles, maximum value, and minimum value (relative fold change "FC" to WT). Statistics: one-way ANOVA followed by Tukey's test.

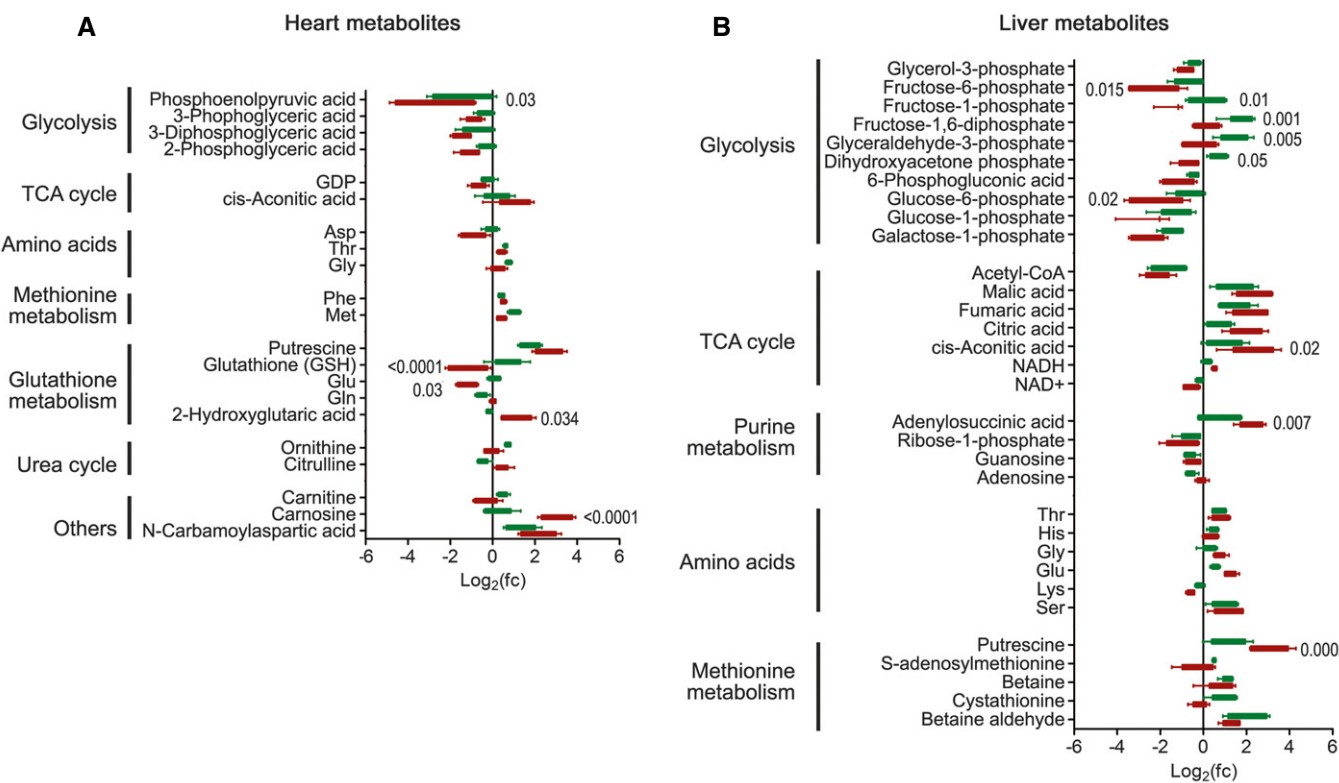

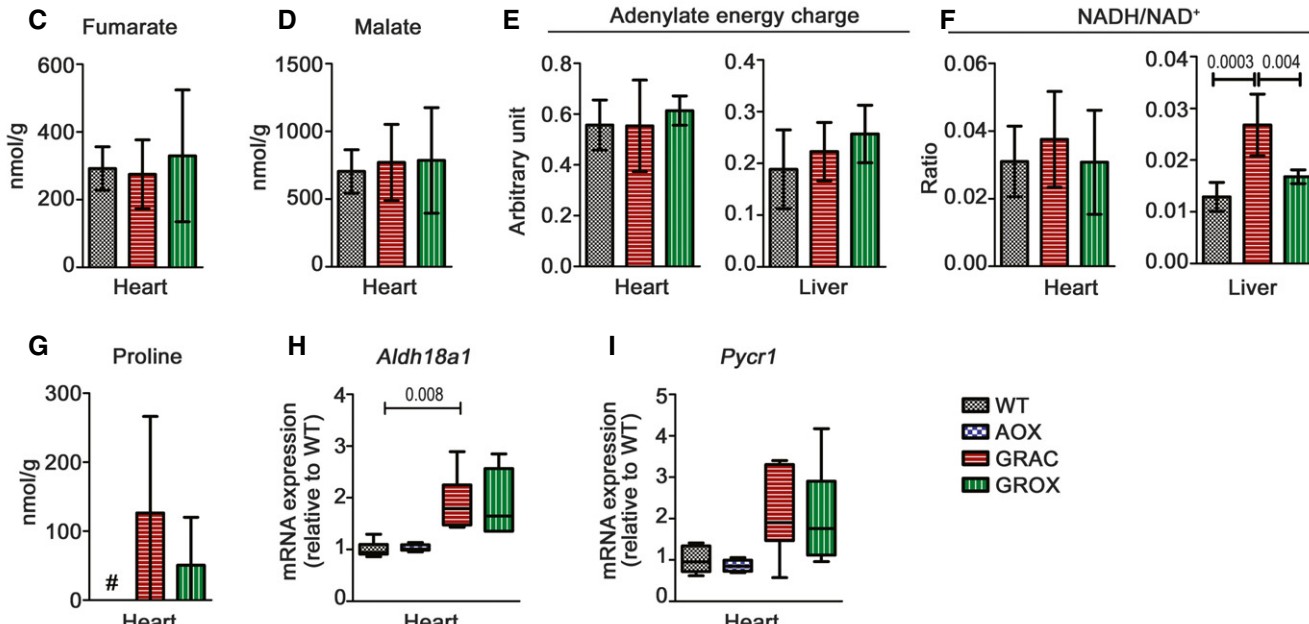

**Figure 5.  AOX has little effect on cardiac energy metabolites but attenuates proline accumulation at onset of disease.**

A    Significantly altered metabolites (FDR < 0.2; $P < 0.05$, $n = 5$/group) in presymptomatic (P150) heart tissue of GRAC (red) and GROX (green) mice.

B    Significantly altered metabolites (FDR < 0.2; $P < 0.05$, $n = 5$/group) in liver tissue of GRAC (red) and GROX (green) mice.

C–F    Concentrations of the TCA cycle intermediates (C) fumarate and (D) malate, (E) adenylate energy charge, and (F) NADH/NAD$^+$ ratio ($n = 5$/group).

G–I    (G) Proline concentration in heart tissue at P200 (# below detection limit) and mRNA expression of proline synthesis-related genes ($n = 6$/group) (H) *Aldh18a1* and (I) *Pycr1* at P150.

Data information: Bar graphs (A–G) represent mean ± SD, and box plots of mRNA expressions represent quartiles, minimum value, and maximum value (relative fold change "FC" to WT). Statistics for graphs (A, B, F, and H): one-way ANOVA followed by Tukey's test.

## AOX restores cardiac mitochondrial respiration

To investigate whether the AOX rescue effect was linked to improved RC assembly and function, we first performed blue native gel electrophoresis (BNGE) analyses. These confirmed the loss of RISP from free cIII dimer ($cIII_2$) and from the cI-$cIII_2$ supercomplex (SC1) in GRAC tissues (Fig 6A). Surprisingly, the amount of this fully assembled $cIII_2$ was significantly increased in the GROX heart when compared to GRAC (Fig 6B and C). In the GROX liver and kidney, the change was opposite, with lower amount of RISP in $cIII_2$ (Fig 6B and C).

Spectrophotometrically measured cIII activity in isolated mitochondria was decreased to < 50% of WT values in all three tissues at P150, a threshold for the appearance of hepatic pathology in juvenile mice (Levéen et al, 2011). Despite that AOX should theoretically not affect cIII activity, it was partially restored in GROX heart mitochondria (Fig 6F), in line with the improved RISP assembly. The activities of cI, cII, and cIV were unchanged in heart and liver (Fig 6D, E and G).

Respirometry confirmed AOX-mediated respiration in all three GROX tissues at P150, as measured using the AOX inhibitor n-propyl gallate (nPG). Respiration did not respond to nPG in AOX mice (with wild-type Bcs1l and normal cIII activity), which indicates that AOX was catalytically active only in the mutants (Fig 6H). The cIII dysfunction did not limit state 3 respiration of liver and kidney mitochondria (Fig 6I and J) when driven by cI-linked substrates (malate, pyruvate, and glutamate to generate NADH), but clearly did so in the heart (Fig 6I). Further stimulation of respiration through cII by addition of succinate revealed cIII deficiency in all three tissues in GRAC mice, with the most severe effect in the heart (Fig 6J). Remarkably, cI- and cII-linked state 3 respiration (malate, pyruvate, glutamate, and succinate at ADP saturation) were rescued to wild-type level in GROX heart mitochondria (Fig 6I and J). A similar effect was found in state 3 respiration of heart and liver at P200 (Fig EV4A and B).

## AOX does not affect ROS damage or defense but normalizes cardiac NO-related gene expression

Limiting excessive ROS production from the RC has been posited as a major mechanism of action of AOX (El-Khoury et al, 2014). To assess ROS production in the affected GRAC tissues, we measured hydrogen peroxide ($H_2O_2$) emission from isolated mitochondria at P150. The Amplex Red-peroxidase assay showed that total $H_2O_2$ emission was decreased in GROX heart and kidney as compared to GRAC (Fig 7A). However, analysis of gene expression related to ROS defense and damage showed that glutathione synthesis and conjugation were upregulated only in the GRAC liver (Fig 7B) and were not affected by AOX. GRAC hearts showed no upregulation of ROS defense at all (Fig 7B) despite the most severe cIII dysfunction. Total tissue glutathione was not changed in GRAC heart or liver but was increased in both GROX tissues (Fig 7C). Mitochondrial aconitase activity, a sensitive marker of ROS damage to iron–sulfur clusters (Yan et al, 1997), and protein carbonylation (Fig EV4C and D) were not significantly different between the genotypes. Urinary isoprostanes, a marker of systemic oxidative stress (Fig 7D), was elevated in both GRAC and GROX. Immunohistochemical staining for the lipid peroxidation product 4-hydroxynonenal was increased

in all three GRAC tissues, but the increase was not prevented by AOX (Fig 7E and F). Finally, we fed the GRAC mice the mitochondria-targeted ROS scavenger mitoQ, which has shown beneficial effects in mouse models of cardiac ischemia–reperfusion injury (Adlam et al, 2005). MitoQ feeding started at P150 had no effect on the survival of the mice by P200 or on their cardiac function (Fig EV1A–F).

Since ROS damage was clearly not instrumental in the cardiac rescue by AOX, we looked at nitric oxide (NO) metabolism that is particularly important in the cardiovascular system and known to interact chemically with ROS and the respiratory chain (Carnicer et al, 2013). We observed a significant increase in NO signaling-related gene expression specifically in GRAC hearts, and these changes were attenuated by AOX (Fig 8A). Decreased mRNA expression of central targets of NO signaling, ryanodine receptor (RyR2), sarco(endo)plasmic reticulum $Ca^{2+}$-ATPase Serca2 (Atp2a2), and phospholamban (Pln) in GRAC heart suggested decreased cardiac contraction-related $Ca^{2+}$ channel function (Fig 8B–D). AOX upregulated these $Ca^{2+}$ channel genes in both Bcs1l mutant and wild-type mice. Expression of endothelial NO synthase (Nos3) was increased in GRAC heart (Fig 8E), and, most strikingly, expression of neuronal NO synthase (nNos, Nos1) was increased over 10-fold in the GRAC heart with a significant attenuation in GROX mice. Western blot analysis confirmed similar upregulation of NOS1 protein (Fig 8F and G). Increased gene expression of dimethylarginine dimethylaminohydrolase 1 (Ddah1), which hydrolyzes the endogenous NOS inhibitors dimethylarginine and monomethylarginine (Fig 8H), and increased product/substrate (citrulline/arginine) ratio (Fig 8I) in GRAC heart suggested increased NO production. However, neither protein nitrotyrosine nor total nitrites were increased in the GRAC hearts (Fig 8J and K).

# Discussion

Here, we show that AOX, a non-mammalian enzyme that can bypass cIII blockade by shunting electrons directly from the quinone pool to oxygen, is able to permanently prevent lethal cardiomyopathy and alleviate multiple other pathologies in cIII-deficient GRAC mice. The GRAC mice, initially bred in C57BL/6JBomTac background (Levéen et al, 2011), have turned out particularly useful in mechanistic and interventional studies (Rajendran et al, 2016; Purhonen et al, 2017, 2018) due to their postnatal symptom-free period and short survival. In the current study, we found that, in the C57BL/6JCrl background, the homozygotes survive to median P210 and develop lethal cardiomyopathy after P150, a manifestations not seen in the GRACILE syndrome patients with early neonatal lethality, but a common manifestation in other mitochondrial disorders. AOX provided a full functional rescue of the cardiomyopathy, restored cI- and cII-linked respiration to wild-type levels, and abrogated the signature of metabolic stress responses. Heart muscle is highly dependent on mitochondrial respiration and operates at constant ATP and phosphocreatine concentrations (Ventura-Clapier et al, 2011; Guzun et al, 2015). Our results clearly indicate that the pathogenesis mechanism leading to cardiomyopathy in GRAC mice is insufficiency of electron flow, resulting in disturbed redox and metabolic homeostasis. Cardiac hypertrophy and a metabolic switch from fatty acid to glucose utilization are hallmarks of

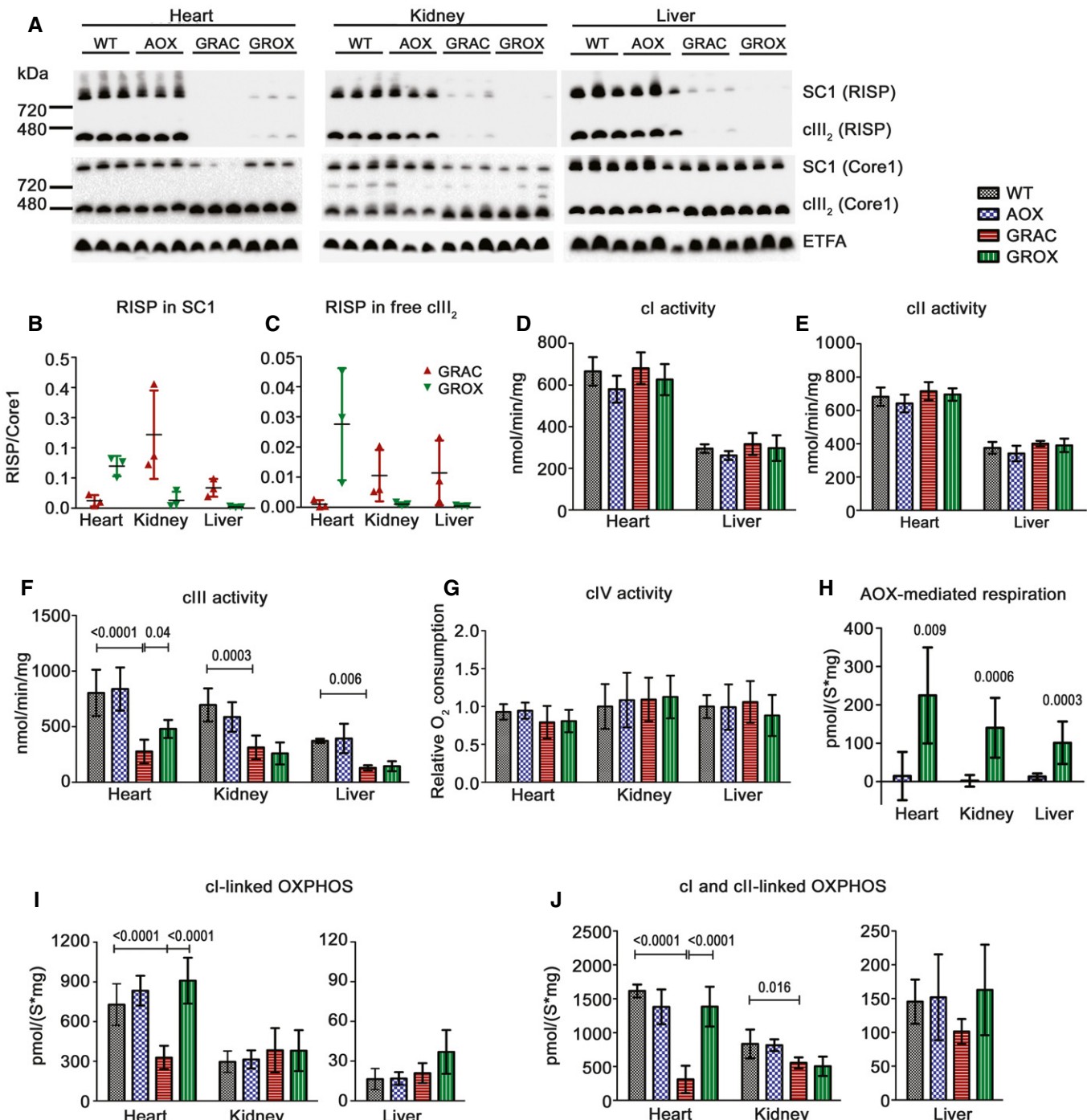

**Figure 6. AOX modifies RISP assembly and rescues mitochondrial respiration in tissue-specific fashion.**

A      Blue native PAGE analysis of RISP and CORE1 assembled into free cIII₂ and supercomplexes (SCs).

B, C    RISP per CORE1 ratio in (B) SC1 and (C) free cIII₂ (n = 3/group).

D–F    Spectrophotometric assay of (D) cI, (E) cII, and (F) cIII activity in isolated mitochondria at P200 (n = 4–6/group).

G      cIV activity as measured using an oxygraph (n = 7/group).

H      n-Propyl gallate-sensitive AOX-mediated cI- and cII-linked state 3 respiration in AOX and GROX mice (n = 7/group, Mann–Whitney U-test).

I, J    cI-linked (I), and cI- and cII-linked (J) state 3 respiration in isolated heart, kidney, and liver mitochondria (n = 8/group). We used malate, pyruvate, and glutamate to generate NADH for the cI. Subsequently, we added the cII substrate, succinate, to obtain cI- and cII-linked respiration in the presence of saturating ADP.

Data information: Bar graphs represent mean ± SD. Statistics: one-way ANOVA followed by Tukey's test. Significant differences between groups (P-value) are indicated on graphs.

Source data are available online for this figure.

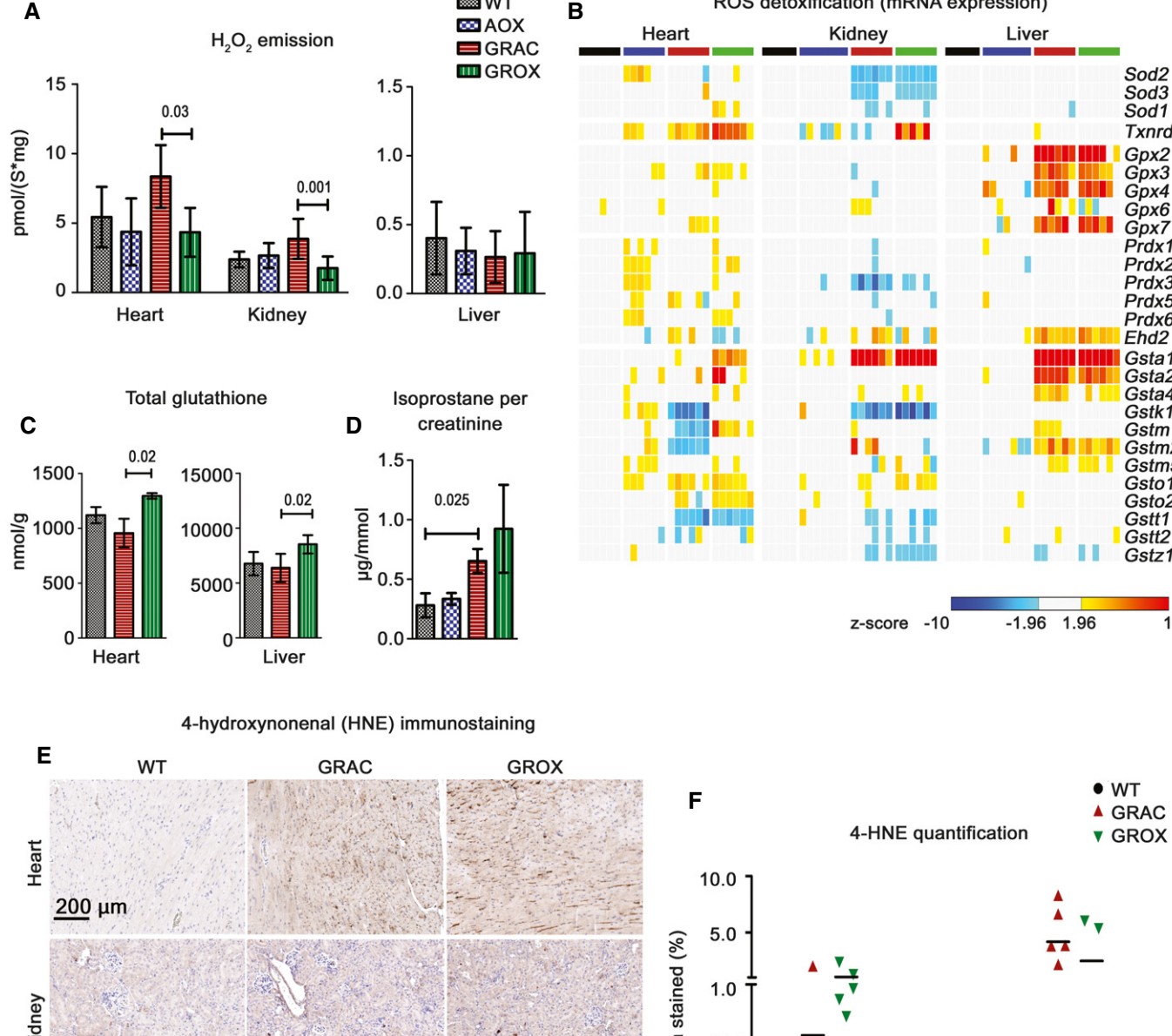

**Figure 7. AOX normalizes ROS production by isolated mitochondria but has little effect on ROS markers in tissues.**

A    Amplex Red-based detection of $H_2O_2$ emission, a surrogate for respiratory chain-derived ROS, during cI- and cII-linked state 3 respiration ($n$ = 6/group).

B    Gene expression related to ROS scavenging.

C    Total glutathione in heart and liver tissue ($n$ = 5/group).

D    Urinary isoprostanes per creatinine as a measure of oxidative stress ($n$ = 4/group).

E, F    Representative images (E) of tissue sections immunostained with an antibody against 4-hydroxynonenal (4-HNE), a lipid peroxidation marker. (F) Quantification of the 4-HNE immunostaining.

Data information: Bar graphs represent mean ± SD. Statistics: Mann–Whitney $U$-test (graph D) and one-way ANOVA followed by Tukey's test (graphs A and C).

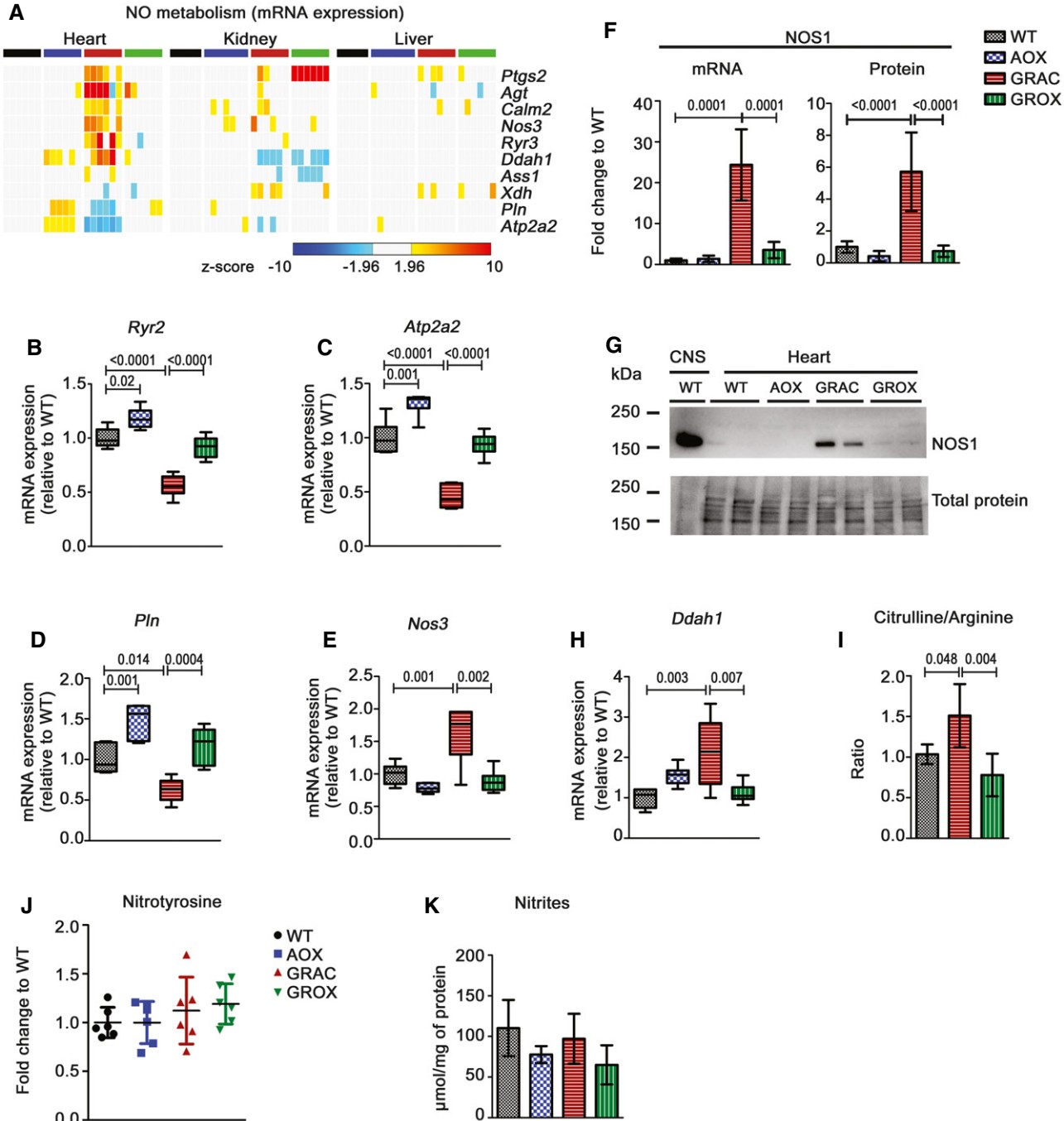

**Figure 8. AOX normalizes cardiac nitric oxide-related gene expression.**

A       Heat map visualization of the expressions of selected nitric oxide (NO)-related genes.

B–E     Gene expression of Ca$^{2+}$ channel genes (B) ryanodine receptor, RYR2, (C) sarco(endo)plasmic reticulum Ca$^{2+}$-ATPase (SERCA2 or ATP2A2), (D) phospholamban (PLN), and (E) endothelial nitric oxide synthase NOS3 ($n = 6$/group).

F, G    Expression of neuronal nitric oxide synthase (NOS1) mRNA (F) and (G) protein with a representative Western blot. WT brain was used as positive control in the first lane (G).

H       Cardiac gene expression of dimethylarginine dimethylaminohydrolase 1 (DDAH1) ($n = 6$/group).

I       Cardiac citrulline-to-arginine ratio ($n = 5$/group).

J       Cardiac protein nitrotyrosine quantified from Western blots ($n = 6$/group).

K       Nitrite concentration in heart tissue ($n = 4$/group).

Data information: Box plots for mRNA expression (B–E and H) represent quartiles, minimum value, and maximum value (relative fold change "FC" to WT). Bar graphs (F, I, and K) and scatter plot (J) represent mean ± SD. Statistics: one-way ANOVA followed by Tukey's test (for graphs B–F, H, and I). Significant differences between groups (*P*-value) are indicated on graphs.

Source data are available online for this figure.

an adaptive response to chronic stress (Hansson *et al*, 2004; Ventura-Clapier *et al*, 2011; Tuomainen & Tavi, 2017), reflected in the GRAC mice by depletion of glycolytic intermediates, downregulation of PPAR-α, and the upregulation of ATF4 (Quiros *et al*, 2017), PGC-1α, and TFAM. These changes were fully or partially prevented in GROX mice, further implying that the underlying metabolic stress was relieved. The surprising partial rescue of cIII assembly and activity by AOX in cardiac mitochondria was similar to what we observed in liver mitochondria upon attenuation of hepatopathy by ketogenic diet (Purhonen *et al*, 2017) and could be secondary to the general improvement in mitochondrial structure and function.

GRACILE syndrome patients and GRAC mice display proximal kidney tubulopathy (Fellman *et al*, 1998; Levéen *et al*, 2011), which we found to progress to kidney atrophy in the older mice. Tubular epithelial cells have a high energy demand, are rich in mitochondria, and thus susceptible to damage from RC dysfunction (Emma *et al*, 2012). The typical atrophic changes in the GRAC kidney were efficiently prevented, and tubular mass was preserved by AOX during the entire life span of the GROX mice. However, the compromised cI- and cII-linked respiration and pathological gene expression changes were not improved in samples prepared from the whole kidney, which suggest that the cIII bypass mainly affected a limited cell population, most likely the tubular epithelial cells.

Surprisingly, AOX provided no long-term histological or functional benefit in the liver. Correspondingly, it did not correct the early-onset systemic manifestation growth restriction and lipodystrophy, which are likely dependent on the liver. Even though *ex vivo* oxygen consumption at saturating substrate concentration showed that AOX was able to support respiration in liver mitochondria, this conferred no functional improvement in liver. One explanation would be that the metabolic threshold for AOX activation was not passed in the liver *in vivo*. Indeed, loss of cIII activity and phosphorylating respiration were not as prominent in the liver as in the heart. The apparent further loss of RISP from cIII in the presence of AOX could simply reflect relaxation of the need to keep cIII assembled and running when AOX replaces its ubiquinol oxidase activity. It is also possible that disturbances in the numerous anabolic and catabolic functions of hepatic mitochondria contribute to the pathogenesis in this tissue, but are not affected by the cIII bypass. Finally, AOX transgene expression was somewhat different in different tissues, as shown in this study and previously (Szibor *et al*, 2017), but this did not correlate with the rescue effect in the tissues we studied. Thus, the disparity in AOX rescue effect was likely due to differences in the severity of the respiration defect in different tissues.

Respiratory chain blockade may lead to increased ROS production via electron leak to oxygen (Murphy, 2009; Brand, 2010). Although loss of RISP inactivates the quinol oxidation site of cIII and thus should decrease rather than increase ROS, cIII dysfunction may lead to ROS production at cI and cII (Korde *et al*, 2011; Sena *et al*, 2013). Indeed, we observed increased ROS production in GRAC heart and kidney mitochondria, and this was prevented by AOX. Increased ROS may cause oxidative damage to proteins and other molecules, which, in turn, should invoke a protective detoxification response. However, expression of glutathione peroxidases and glutathione-S-transferases was induced only in the GRAC liver. In the heart and kidney, apart from locally increased lipid peroxidation, there were no consistent signs of ROS damage or detoxification

response. The lack of transcriptional response to oxidative damage was particularly striking in the GRAC heart, ruling this out as a significant early pathogenetic mechanism, although it may still play a role in end-stage disease. Dogan *et al* (2018) recently reported that AOX expression in a *Cox15* knock-out model of cIV deficiency exacerbated myopathy and decreased survival. Harmful perturbation of redox signaling via AMPK/PGC-1α was suggested as a mechanism. Interestingly, AOX did not worsen any of the multiple visceral and systemic phenotypes we thoroughly investigated in the cIII-deficient GRAC mice. Neither did we observe worsening in any behavioral or motor activity parameters (our behavioral scoring and additional unpublished phenotyping data from the German Mouse Clinic) that could have suggested development of skeletal myopathy, not otherwise present in GRAC mice, upon AOX expression. Moreover, feeding GRAC mice the mitochondria-targeted antioxidant mitoQ had no effect on the progression of cardiomyopathy. Compound heterozygous or homozygous missense mutations in *COX15* have been identified in patients presenting with cardiomyopathy and/or encephalopathy, but not skeletal myopathy (Alfadhel *et al*, 2011). Whether the findings of Dogan *et al* (2018), showing adverse effect of AOX expression in the mice with Cre-mediated skeletal muscle-specific *Cox15* knock-out, apply to other models of cIV deficiency and/or mitochondrial myopathy remains to be investigated. Nevertheless, given that we found obvious differences in the response to AOX-mediated cIII bypass in different tissues in our model, it is possible that the responses would be disparate in other models, depending on the exact nature of the mutation and respiration defect.

We found that NOS1, an enzyme generating NO, a major modulator of cardiac function and calcium fluxes (Pechanova *et al*, 2015), was highly upregulated specifically in the GRAC heart, and this was to large extent prevented in GROX hearts. NOS1 and NOS3 upregulation was accompanied by changes in NO- and $Ca^{2+}$ channel-related gene expression, both alleviated by AOX. NO itself is relatively inert but reacts rapidly with superoxide, the primary ROS emitted by the RC, forming highly reactive and cytotoxic peroxynitrite (Pechanova *et al*, 2015). Peroxynitrite can impair cardiac function via multiple mechanisms, including inhibition of cI and cIV by S-nitrosation and nitrosylation, respectively (Burwell & Brookes, 2008; Chouchani *et al*, 2013; Pechanova *et al*, 2015), which in the case of GRAC mice could exacerbate the cardiomyopathy via a vicious circle-type mechanism. However, cIV activity was similar in all groups and lack of increased protein tyrosine nitration and total nitrates suggests NOS uncoupling or loss of function despite highly increased expression, rather than increased NO production in the GRAC heart. These findings imply that impaired NO metabolism should be further investigated as a pathological process in mitochondrial cardiomyopathy.

In summary, our findings represent the first proof-of-concept that RC bypass can alleviate pathological manifestations in a genetic mouse model of a human mitochondrial disorder. While restoring (e.g., viral delivery) or correcting (e.g., CRISPR/Cas9 genome editing) the mutated gene in the affected tissue is theoretically the most efficient treatment strategy in monogenic diseases, and has shown promise in animal models (Logan *et al*, 2014; Ohmori *et al*, 2017), this requires gene-specific tools and is only under development for mtDNA mutations. Therefore, as an enzyme that can potentially alleviate cIII or cIV blockade due to a wide variety of mutations, AOX is worth further preclinical investigations, e.g., using viral

delivery to the affected tissues in mouse models. Such experiments should shed light on the possible immunological adverse effects associated with the introduction of a non-mammalian protein into juvenile or adult tissues. Alternatively, there is suggestive evidence that beneficial cIII bypass-like effects can be achieved pharmacologically, for example with vitamin $K_2$ (Vos et al, 2012) or methylene blue (Atamna et al, 2008). Our results urge the search for novel pharmacological bypass strategies potentially translatable to patients. Finally, the AOX transgenic mice provide an excellent tool to dissect the mechanisms underlying the wide variety of manifestations in mitochondrial disorders, the understanding of which is a prerequisite for the development of novel therapies.

# Materials and Methods

### Animal breeding and husbandry

Heterozygous knock-in ($Bcs1l^{c.A232G}$) mice in the C57BL/6JCrl (RRID: IMSR_JAX:000664) background and AOX transgenic mice (Szibor et al, 2017) were maintained in the animal facilities of University of Helsinki, Finland. All mice were housed in individually ventilated cages with 12-h light/12-h dark cycle at a temperature of 22–23°C, and water and food (2018 Teklad global 18% protein rodent diet, Envigo) available ad libitum. The animal studies were approved by the animal ethics committee of the State Provincial Office of Southern Finland (ESAVI/6142/04.10.07/2014 and ESAVI/6365/04.10.07/2017) and were performed according to FELASA (Federation of Laboratory Animal Science Associations) guidelines.

$Bcs1l^{c.A232G}$ homozygotes survive to over P150 in the C57BL/6JCrl background (Purhonen et al, 2017). The two mouse strains were crossed to generate double heterozygous mice, which were backcrossed to the C57BL/6JCrl background for several generations. Wild-type or $Bcs1l$ heterozygous animals, littermates whenever possible, were used as wild-type (WT) controls. Mice carrying a single copy of the AOX transgene (AOX mice), littermates whenever possible, were used as a second control group. Both genders were used in the experiments and were not separated unless separate analysis for males and females is indicated in Results. Genotyping for the $Bcs1l$ mutation and AOX transgene was performed as described (Levéen et al, 2011; Szibor et al, 2017).

Mouse health was monitored by manual behavioral scoring and weighing according to the ethical permit and as described elsewhere (Rajendran et al, 2016; Purhonen et al, 2017). The time points for assessments were chosen to verify early and late-onset manifestations: growth and survival data from weaning on, whole-body metabolism and DEXA data between 10 and 14 weeks of age (at German Mouse Clinic), presymptomatic cardiac data at 5 months (P150), end-stage disease for GRAC mice at 6–7 months (P200), and survival and tissue histology of surviving GROX mice at up to 22 months (P680). The total number of experimental mice used in this study was 301.

### Assessment of body composition and whole-body metabolism

At the German Mouse Clinic (GMC), the mice were maintained from 8 weeks of age until sacrificing at week 21 according to the GMC housing conditions (www.mouseclinic.de) and German laws. All tests

were approved by the responsible authority of the Regierung von Oberbayern. Twenty mice per genotype (WT, AOX, GRAC, and GROX), 10 mice per gender, were phenotyped between ages P56 and P112. The determination of energy expenditure by indirect calorimetry was performed at P77. Dual-energy X-ray absorptiometry (DEXA) was used to measure bone density, fat mass, and lean mass at P98.

### Salt provocation and antioxidant administration

Powdered chow (2018 Teklad global 18% protein, Envigo) was supplemented to contain 6% NaCl (w/w), moistened, and manually compressed into pellets. The pellets were air dried for 24 h and stored frozen. Control diet was prepared similarly but without the addition of NaCl. The NaCl-supplemented chow was administered ad libitum from P150 onwards. For antioxidant feeding, drinking water was supplemented with 0.50 mM mitoquinol mesylate (mitoQ, a kind gift from MitoQ Ltd, New Zealand) starting from P150.

### Echocardiography

The cardiac dimensions and ejection volume of mice were measured by using Vevo 2100 imaging system (Fujifilm). B-mode and M-mode were used to measure left ventricle dimensions, and the average values were used for analysis.

### Clinical chemistry and urine analysis

Alanine aminotransferase and alkaline phosphatase were measured from plasma, and albumin and creatinine from urine samples using Siemens ADVIA1650 analyzer. For urine collection, the mice were kept in metabolic cages for 24 h under 12-h/12-h light/dark cycle. Other urine analyses were performed by pipetting 15 μl of urine per each test area on human urine strips (Combur10 Test UX, Roche), followed by reading of the values in Urisys 1100 urine analyzer (Roche).

### Sample collection

For end point sacrificing and sample collection, the mice were not fasted for longer than 2–3 h due to the hypoglycemia associated with the mutant phenotype. All samples were collected at the same time of the day during the light period of the mice. The mice were anesthetized with pentobarbital, blood was collected into Li-heparin tubes by cardiac puncture, and plasma was separated. Organs were immediately dissected, and snap-frozen in liquid nitrogen or immersed in 10% buffered formalin solution. Approximately 1 mm$^3$ pieces of tissue were fixed in 1.5% glutaraldehyde and 1.5% paraformaldehyde in 0.1 M Sörensen buffer pH 7.2 for electron microscopy. Brains ($n = 6$ per genotype at P200; $n = 8$ for AOX and $n = 10$ for GROX at P600) were immersion-fixed in 4% paraformaldehyde in 0.1 M sodium phosphate buffer pH 7.4 for 48 h followed by cryoprotection in 30% sucrose/0.05% sodium azide in 50 mM Tris-buffered saline (TBS). Frozen coronal sections (40 μm) were cut through cerebrum, while cerebella were cut sagittally and stored in cryoprotectant solution as described (Tegelberg et al, 2017). Blood glucose and lactate levels were measured using commercial quick sticks and meters (Freestyle lite and Lactate Pro

LT-1710). In living animals, the ventral tail artery was punctured under light isoflurane anesthesia to measure glucose and lactate.

## Histology and electron microscopy

Tissue sections were stained with standard methods to assess general histology (hematoxylin–eosin) and fibrosis (Sirius Red). The area stained by Sirius Red was quantified using ImageJ software (National Institutes of Health, Bethesda, MD, USA) from 5 to 10 random fields. Tissues were prepared for transmission electron microscopy using standard methods and examined in a Philips/FEI CM 100 BioTWIN transmission electron microscope (Hillsboro, OR) at 60 kV. Images were captured in side-mounted Olympus Veleta camera (Center Valley, PA) with a resolution of 2,048 × 2,048 pixels (2k × 2k).

## Immunohistochemistry

One-in-twelve series of free-floating cryosections were stained as previously described (Bible *et al*, 2004). Primary antibody against glial fibrillary acidic protein (GFAP: Z0334, DAKO, Agilent Technologies) diluted 1:4,000 in 10% normal serum in TBS-T was incubated overnight at 4°C. Standard paraffin sections were immunostained with primary antibodies against Ki67, cleaved caspase-3, and 4-HNE (see Appendix Table 3), using Vectastain Elite peroxidase reagents (Vector Laboratories) and standard protocols.

## Western blot analyses

Frozen tissues were homogenized in RIPA buffer (50 mM Tris, pH 8, 150 mM $NaCl_2$, 1% Triton X-100, 0.5% sodium deoxycholate, 0.1% SDS) with protease inhibitors (Complete mini, Roche). Protein concentration was measured using Bradford reagent (Bio-Rad). Approximately 10 μg of protein per lane was run on Mini-PROTEAN StainFree© precast gradient gels (Bio-Rad). StainFree© dye was activated by UV light for one minute and proteins transferred to PVDF membrane using Trans-Blot Turbo Transfer System (Bio-Rad). The filters were then imaged for total protein, blocked with 5% milk powder in TBST, and stained with primary antibodies (Appendix Table S3). Appropriate HRP-conjugated secondary antibodies and enhanced chemiluminescence were used for the detection. Band intensities, as recorded with ChemiDoc MP digital imager (Bio-Rad), were normalized to total protein.

## Isolation of mitochondria

Liver and kidney samples were minced with scissors and then homogenized with a teflon-glass potter homogenizer containing homogenization buffer (225 mM mannitol, 75 mM sucrose, 10 mM Tris–HCl, 1 mM EGTA, 1 mg/ml albumin, and pH set to 7.4 at +4°C). For respirometry, a crude mitochondrial fraction was obtained by a two-step differential centrifugation (10 min at 800 *g* and 10 min at 7,800 *g*). For other assays, differential centrifugation was continued, including a 19% Percoll solution (GE Healthcare), and the mitochondria were stored as pellets at −80°C. Heart mitochondria were isolated similarly, but the tissue was minced in trypsin–EDTA solution (Thermo Scientific). Trypsin was inactivated with fetal bovine serum and removed before homogenization. All steps were carried at 0–4°C.

## Blue native gel electrophoresis

Digitonin-solubilized mitochondria were prepared and subjected to BNGE as described (Davoudi *et al*, 2014).

## Respirometry

Oxygen consumption by isolated mitochondria was measured polarographically in Mir05 buffer (110 mM sucrose, 60 mM lactobionic acid, 20 mM taurine, 20 mM HEPES, 10 mM $KH_2PO_4$, 3 mM $MgCl_2$, 0.5 mM EGTA, and 1 g/l fatty acid free bovine serum albumin, pH 7.1) in a 2-ml chamber (OROBOROS Instruments). Sample, substrates, inhibitors, and uncouplers were injected in following order: (i) 1 mM malate, 5 mM pyruvate and 5 mM glutamate; (ii) sample; (iii) 1.25 mM ADP; (iv) 10 μM cytochrome c; (v) 10 mM succinate; (vi) 100 μM propyl gallate; (vii) 1 μg/ml oligomycin A; (viii) carbonyl cyanide 4-(trifluoromethoxy) phenylhydrazone (FCCP) titration to maximum respiration; (ix) 0.5 μM rotenone; and (x) 1 μg/ml antimycin A. Complex IV activity was measured using 2 mM ascorbate and 0.5 mM N,N,N′,N′-tetramethyl-p-phenylenediamine (TMPD) as substrates. Chemical and oxygen-dependent TMPD auto-oxidation was determined after addition of 10 mM sodium azide.

## Mitochondrial $H_2O_2$ emission

The oxygraph was equipped with a fluorometer and set for Amplex Red/horseradish peroxidase (HRP) assay to measure mitochondrial $H_2O_2$ emission simultaneously with respirometry (Makrecka-Kuka *et al*, 2015). To validate the assay, HRP-independent and catalase-insensitive conversion of Amplex Red to the fluorescent resorufin was initially tested. Liver and kidney, but not heart, mitochondria possessed high rate of HRP-independent artificial Amplex Red oxidation due to presence of mitochondrial carboxylesterases (Miwa *et al*, 2016). This signal was inhibited by adding 100 μM phenylmethylsulfonyl fluoride. Final assay conditions in Mir05 buffer were 10 μM Amplex Ultra Red (Invitrogen), 1 U/ml HRP, 6 U/ml superoxide dismutase, 100 μM PMSF (for liver and kidney), and substrates for cI- and cII-linked respiration as described in the previous chapter. The assay was calibrated with a 0.4 nmol bolus of $H_2O_2$ in every respiratory state for which results are reported. As a background for heart and kidney mitochondria, the rate of fluorescence was recorded when mitochondria were kept without any substrates. For liver mitochondria, background was taken as a catalase-insensitive (700 U/ml) fluorescence flux.

## Enzyme activity and lipid peroxidation assays

The activity of cIII in isolated mitochondria was measured using a spectrophotometric assay as described previously (Davoudi *et al*, 2014). Aconitase activity (MAK051, Sigma) and urinary isoprostanes (EA85, Oxford Biomedical Research) were measured using the indicated commercial kits. Carbonylated proteins were labeled on-filter with 2,4-dinitrophenylhydrazine (phosphoric acid solution, 42215, Sigma-Aldrich) and detected with anti-dinitrophenol antibody (Oxidized Protein Western Blot Kit, ab178020, Abcam) as described (Colombo *et al*, 2016). Nitrites in heart tissue were measured using Griess reagent (03553, Sigma-Aldrich).

Isoprostanes were quantified from urine using a commercial ELISA kit (ab175819, Abcam).

## Transcriptomics

For RNA sequencing, rRNA was removed and the sequencing libraries were prepared using Illumina TruSeq Stranded Total RNA Library Prep Kit with Ribo-Zero Human/Mouse/Rat. Libraries were sequenced on Illumina NextSeq 500 instrument using 75 bp kit. Adapter sequences and low-quality reads were removed from the data using cutadapt (Martin, 2011), and data were further screened for remaining rRNA reads using SortMeRNA (Kopylova et al, 2012). The data were mapped to *M. musculus* genome GRCm38.p4 using STAR (Dobin et al, 2013). The original genome sequence and annotation were augmented with sequence and annotation for AOX. Count data were processed in R using GenomicFeatures and GenomicAlignments (Lawrence et al, 2013), and the differential expression analysis was carried out using DESeq2 (Love et al, 2014). PCA plots were generated using ClustVis online tool (www.b iit.cs.ut.ee/clustvis).

## Pathway enrichment analysis

Enrichment analyses were performed using filtered gene sets ($|FC| > 1.5$, $P < 0.05$) against Reactome database using www.mouse mine.org. To visualize pathway changes, heat maps were generated by Gitool.2.3.1 with WT mean-centered $Z$-score values and gene mapping for selected pathways as described (Gundem & Lopez-Bigas, 2014; Perez-llamas & Lopez-Bigas, 2011). Pathway heat maps were generated based on $Z$-score, showing upregulation (brown color) and downregulation (blue color) of each pathway. $P$-values for group comparison (labeled 1, 2, and 3) were calculated using one-way ANOVA followed by selected comparisons and the Benjamin–Hochberg FDR correction.

## Metabolomics

Targeted quantitative metabolomics covering 116 metabolites was performed using capillary electrophoresis–mass spectrometry (CE-TOMFS and CE-QqQMS) at Human Metabolome Technologies Inc., Japan (http://humanmetabolome.com/en/). Details of the protocol are available upon request. We omitted the AOX group in the metabolomics since ectopic AOX was shown to be inert in healthy wild-type mice (Szibor et al, 2017).

## Statistics (excluding transcriptomics)

For normally distributed data, one-way ANOVA followed by Tukey's test or by selected comparisons using *t*-tests with Welch's correction was used. Normality of data was assessed by the d'Agostino–Pearson omnibus normality test, and equality of variances by Bartlett's test. Data not compatible with parametric tests were assessed by Kruskal–Wallis and Mann–Whitney *U*-tests. Survival curves were analyzed using log-rank Mantel–Cox test. GraphPad Prism 7 software (GraphPad Software Inc., La Jolla, CA, USA) was used for the statistical analyses. Group sizes (*n*) and the statistical tests used are described in figure legends, and exact *P*-values are shown in the graphs.

### The paper explained

**Problem**

Mitochondrial diseases are genetic disorders of energy metabolism that can affect any one or several organs of the body, including skeletal muscle, heart, brain, and visceral organs. Nowadays, early diagnosis of mitochondrial disease is often possible using modern molecular genetics, but treatment options remain scarce. Mutations in the *BCS1L* gene are the most common cause of mitochondrial diseases affecting the respiratory chain complex III (cIII). GRACILE syndrome (Growth Restriction, Aminoaciduria, Cholestasis, liver Iron overload, Lactic acidosis, and Early death), with neonatal lethality, is the most severe of them. Due to the strikingly similar disease in all GRACILE syndrome patients, Prof. Fellman's group previously introduced the underlying missense mutation (*Bcs1l*$^{c.A232G}$, *Bcs1l*$^{p.S78G}$) into mice. This mouse model faithfully recapitulates most of the symptoms of the patients. In search for novel strategies to alleviate cIII deficiency-related pathology, we crossed the *Bcs1l*$^{c.A232G}$ mice with transgenic mice expressing alternative oxidase (AOX), a mitochondrial inner membrane enzyme that can bypass respiratory electron flow blockade when the quinol oxidation capacity of cIII is compromised.

**Results**

Mice homozygous for the *Bcs1l*$^{c.A232G}$ mutation survived approximately 200 days and died of dilating cardiomyopathy, a novel late-onset phenotype in the C57BL/6JCrl background. In contrast, the homozygotes carrying the AOX transgene lived approximately 600 days and never developed the cardiomyopathy. AOX also ameliorated the severe kidney disease and focal astrogliosis of the brain in the homozygotes. Surprisingly, AOX did not correct the liver disease, poor growth, and loss of white fat, suggesting different disease mechanism in different tissues. AOX corrected the abnormal ultrastructure of mitochondria and mitochondrial respiration in those tissues, in which the tissue pathology was alleviated. Heart and kidney mitochondria from the homozygotes showed elevated reactive oxygen species (ROS) production, but analyses of ROS damage suggested that the beneficial effects of AOX were not ROS-related. Instead, we found that AOX normalized cardiac gene expression related to nitric oxide metabolism and signaling, a major modulator of cardiovascular functions. We conclude that AOX efficiently prevented tissue pathology in the *Bcs1l* mutant mouse model of cIII deficiency by restoring respiration, preferably in the most highly oxidative tissues such as heart.

**Impact**

Our study in the patient mutation knock-in model of cIII deficiency is the first proof-of-concept that bypass of the cIII-cIV segment of the respiratory electron transfer can alleviate pathological manifestations in a physiologically relevant genetic mouse model of a human mitochondrial disorder. These findings highlight the potential of AOX as a tool to unravel disease mechanisms, and also urge further studies of AOX in preclinical models, e.g., using viral delivery to the affected tissues in mouse models, as well as studies to search for novel pharmacological bypass strategies potentially translatable to patients.

## Data availability

Additional raw data and detailed protocols are available from the authors upon request. Transcriptomics data have been deposited to ArrayExpress database at EMBL-EBI under the accession ID E-MTAB-7416 (https://www.ebi.ac.uk/arrayexpress/experiments/E-MTAB-7416).

**Expanded View** for this article is available online.

## Acknowledgements

We thank Prof. Hannu Sariola for expert assistance with tissue histopathology; Dr. Eric Dufour for advice on respirometry; Praveen Dhandapani for the AOX genotyping protocol; Elisa Altay, Päivi Leinikka, and Nada Bechara-Hirvonen for expert technical assistance; the staff of the DNA Sequencing and Genomics Laboratory, University of Helsinki, for running the RNAseq; the staff of the BioEM Lab (at C-Cina), Biozentrum, University of Basel, the Core Facility for Integrated Microscopy, Panum Institute, University of Copenhagen, and Ola Gustafsson of Microscopy Facility at the Department of Biology, Lund University, for providing the electron microscopy facilities and assistance; and finally Dr. Sanna Marjavaara for helpful initial discussions. This study was supported by grants from Academy of Finland (grant 259296 to VF, 256615 and 272376 to HTJ), Swedish Research Council (grant 521-2011-3877 to VF), European Research Council (Advanced Grant 232738 to HTJ), Finska Läkaresällskapet (to VF), Foundation for Pediatric Research in Finland (to VF), Folkhälsan Research Center (to VF, JK), and German Federal Ministry of Education and Research (Infrafrontier grant 01KX1012 to MHdA).

## Author contributions

HTJ, MS, VF, and JK invented the concept to combine the mouse models; HTJ and MS provided the mouse strain with broad AOX expression; VF and JK group combined the strains; JRa, JP, ST, EM, VF, and JK conceived and designed the experiments; HF, VG-D, and MHA conceived and supervised phenotyping experiments at the GMC; JRa performed the animal experiments (echocardiography, metabolic sample collection, sickness scoring, tissue sampling); JRa performed the laboratory analyses (histology, protein analyses, BNGE, respirometry, cIII activity, metabolomics and transcriptome pathway enrichment analysis, ROS markers); JP performed respirometry, histology, immunohistochemistry quantitation, and Amplex Red assay analyses; ST performed brain histology and immunohistochemistry analyses; O-PS and PA performed transcriptomics analysis; JK performed immunohistochemistry and ROS marker analyses; MM performed electron microscopy analyses; JRo and HF performed indirect calorimetry and DEXA analysis; JRo and HF analyzed the data from GMC; JRa, JP, ST, VF, and JK analyzed the data from Helsinki; EM, HTJ, and MS participated in interpretation of the results; JRa, JP, VF and JK wrote the manuscript draft; all authors revised the manuscript and have contributed substantially to the work reported.

## Conflict of interest

MS is a shareholder of a commercial enterprise that is dedicated to developing therapeutics based on AOX. The authors declare that they have no conflict of interest.

## For more information

(i)   GRACILE syndrome: https://www.omim.org/entry/603358.

(ii)  *BCS1L* gene: https://www.omim.org/entry/603647.

(iii) United Mitochondrial Disease Foundation (UMDF): https://www.umdf.org/.

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
