## [Review Process File · EMBO Molecular Medicine]

Alternative oxidase-mediated respiration prevents lethal mitochondrial cardiomyopathy

Jayasimman Rajendran, Janne Purhonen, Saara Tegelberg, Olli-Pekka Smolander, Matthias Mörgelin, Jan Rozman, Valerie Gailus-Durner, Helmut Fuchs, Martin Hrabe de Angelis, Petri Auvinen, Eero Mervaala, Howard T. Jacobs, Marten Szibor, Vineta Fellman, Jukka Kallijärvi

Review timeline:

Submission date:	8 August 2018
Editorial Decision:	21 September 2018
Revision received:	16 October 2018
Editorial Decision:	26 October 2018
Revision received:	8 November 2018
Accepted:	12 November 2018

Editor: Lise Roth

Transaction Report:

1st Editorial Decision

21 September 2018

Thank you for the submission of your manuscript to EMBO Molecular Medicine. We have now heard back from the three referees whom we asked to evaluate your manuscript.

As you will see from the reports below, the three referees are positive and support publication of the article in EMBO Molecular Medicine pending appropriate revisions. Addressing the reviewers concerns in full will be necessary for further considering the manuscript in our journal. Particular attention should be given to the discussion, which should address the organ-specificity of the effects and the potential application to human disease. EMBO Molecular Medicine encourages a single round of revision only and therefore, acceptance or rejection of the manuscript will depend on the completeness of your responses included in the next, final version of the manuscript.

EMBO Molecular Medicine has a "scooping protection" policy, whereby similar findings that are published by others during review or revision are not a criterion for rejection. Should you decide to submit a revised version, I do ask that you get in touch after three months if you have not completed it, to update us on the status. Please also contact us as soon as possible if similar work is published elsewhere. If other work is published, we may not be able to extend the revision period beyond three months.

I look forward to receiving your revised manuscript.

***** Reviewer's comments *****

Referee #1 (Comments on Novelty/Model System for Author):

mouse models sometime do not reflect human diseases but - there are no other relevant models

Referee #1 (Remarks for Author):

This interesting and extensive manuscript -shows the ability of ectopic AOX expression to alleviate mitochondrial dysfunction in a mouse model of mitochondrial complex III deficiency. The findings corroborate previous studies performed in *Drosophila* models with complex IV defects. The work is thorough, and the data are ample to support the conclusions. I have only minor comments mainly related to linking human disease to the present findings, as follows.

Introduction:

1) For the benefit of the reader-it would be nice to have a visual scheme/figure pointing out the location/function of CIII and AOX in electron flow/OXPHOS

2) The meaning of "different genetic background from previous " -page 5- is unclear
Does the GRAC mouse better reflect human disease than the other models?

Results:

1) Were plasma /urine metabolites measured? to compare metabolic GRAC results with patients with CIII and other mito/metabolic diseases (obviously in patients tissue metabolites are usually not measured) ;For example: Plasma -Proline and Alanine are often elevated in Lactic academia- could this be the reason for elevated Proline (inhibited proline oxidase by lactate- see also discussion) ;plasma Tyrosine and Methionine and plasma tyrosine-metabolites are often secondary elevated with liver dysfunction

2) What is - "CI and CII-linked OXPHOS-state 3 respiration"- fig 6 ? Explain)pyruvate+succinate substrates ?(

3) Normal wt or AOX values for comparison are lacking in some figures; 5g, 6b/c

4) Could elevated NO -inhibit CIV activity and thereby OXPHOS in situ ?

Discussion -

1) Re tissue specificity; compared wt with GROX- could there be some sort of threshold value that is crucial for liver function? i.e. liver has a priory low activities and is therefore more vulnerable ? What is presently known human/animal models?

2) Elaborate on the possibility if /how AOX could be delivered for example by a viral vector to patients?

Referee #2 (Comments on Novelty/Model System for Author):

This is a very interesting manuscript with relevant pre-clinical data.

Referee #2 (Remarks for Author):

Rajendran et al describes the results of expression of *Ciona intestinalis* alternative oxidase (AOX) in a mouse model of Complex III deficiency. Specifically, this mouse model is a knock-in for the homozygous *Bcs1l*c.232A>G (*Bcs1l*p.S78G) mutation associated with the GRACILE syndrome and recapitulates many of the clinical manifestations, including growth failure, progressive hepatopathy, kidney tubulopathy, and short survival in a C57Bl/6NBomTac background. In this paper the Authors used another genetic background (C57BL/6JCrI) in which the homozygotes for the *Bcs1l*.S78G mutation, survive longer and develop additional later onset phenotypes to test the effects of AOX expression.

The manuscript is well written with interesting data. I have few comments:

-In Figure 1 d-i the histograms show different data obtained both in male and female. However, the body weight is reported only for males. Is there any difference in female?

-On page 8 this sentence: "AOX had only a minor effect on the hepatic metabolites (Fig. 4b)" refers to figure 4c and not on 4b as indicated.

-The Authors demonstrated that AOX expression rescue pathological phenotype mainly in heart and kidney but not liver. The data reported in figure 6a show that RISP in SC1 and CIII is higher in GRAC vs GROX in kidney and liver, but the enzymatic activity of Complex III is comparable between heart and kidney, but significantly reduced in liver. Is it possible that AOX can't rescue the phenotype of the liver because there is a partially assembled CIII, although not biochemically active? Moreover, western-blot analysis in supplementary figure 4 show that AOX expression is the lowest in liver and kidney. Could this also contribute to the lack of rescue in the liver?

-I would strongly recommend adding in the discussion the recently published paper by Dogan SA et al "Perturbed Redox Signaling Exacerbates a Mitochondrial Myopathy", Cell metabolism, in view of possible human therapeutic strategy based on AOX expression

Referee #3 (Remarks for Author):

A very well performed study proposing a natural method to replace defective mitochondrial respiration. The authors propose different damage mechanisms between tissues (energetics vs. ROS damage) to explain different responses to AOX. I would like to see data on transgene expression in different tissues because this could be an alternative explanation.

1st Revision - authors' response

16 October 2018

Referee #1 (Remarks for Author):

This interesting and extensive manuscript -shows the ability of ectopic AOX expression to alleviate mitochondrial dysfunction in a mouse model of mitochondrial complex III deficiency. The findings corroborate previous studies performed in Drosophila models with complex IV defects. The work is thorough, and the data are ample to support the conclusions. I have only minor comments mainly related to linking human disease to the present findings, as follows.

Introduction:

1) For the benefit of the reader-it would be nice to have a visual scheme/figure pointing out the location/function of CIII and AOX in electron flow/OXPHOS

Thank you for the suggestion. We have now prepared a cartoon depicting the respiratory chain complexes, AOX and changes in electron flow when cIII is compromised. (**Synopsis Figure**).

2) The meaning of "different genetic background from previous" -page 5- is unclear
Does the GRAC mouse better reflect human disease than the other models?

The GRAC mice faithfully recapitulate most manifestations of the GRACILE syndrome, e.g. growth restriction, liver and kidney disease, and early death, in all three genetic backgrounds that have been studied over the years (mixed 129Sv:C57BL/6JBomTac, congenic C57BL/6JBomTac and congenic C57BL/6JCrI). There are differences mainly in the survival in the different colonies. In a C57BL/6JBomTac-derived colony, the GRAC mice have a short survival of 35 days due to a lethal metabolic crisis (Kotarsky et al, 2012; Leveen et al, 2011; Rajendran et al, 2016). For the current study, we transferred the mice to another facility and used a slightly different congenic background, C57BL/6JCrI. On this background, the disease progression is somewhat milder and the mice do not succumb to the early metabolic crisis. Their extended survival, to up to 200 days (Purhonen et al, 2017) brings additional phenotypes, most prominently the lethal late-onset cardiomyopathy, which has not been reported in the patients and in mice with short survival. Therefore, GRAC mice in all studied backgrounds reflect human cIII deficiency, but the main focus of this study is the late-onset phenotypes, mainly the cardiomyopathy. We have clarified this in the text (**PAGE 3, LINES 5**)-62).

Results:

1) Were plasma /urine metabolites measured? to compare metabolic GRAC results with patients with CIII and other mito/metabolic diseases (obviously in patients tissue metabolites are usually not measured) ;For example: Plasma -Proline and Alanine are often elevated in Lactic academia- could this be the reason for elevated Proline (inhibited proline oxidase by lactate- see also discussion)

plasma Tyrosine and Methionine and plasma tyrosine-metabolites are often secondary elevated with liver dysfunction.

For this study, we measured liver enzymes from plasma and albumin, creatinine and isoprostanes from urine. However, we reported changes of plasma metabolites in GRAC mice in our previous study (Purhonen et al, 2017), in which we show that plasma alanine and proline were, along with general elevation of plasma amino acids, significantly elevated in GRAC mice at P95. Here, we show that proline is highly increased in heart tissue. The referee is right that the elevated proline level may be linked to elevated lactate, which is a known inhibitor of proline oxidase, as shown first by Kowaloff et al. (1977). We have now cited this paper. **(PAGE 8, LINES 174-175)**

2) What is - "CI and CII-linked OXPHOS-state 3 respiration"- fig 6 ? Explain) pyruvate+succinate substrates ?

CI&cII-linked respiration is a respiratory state in which a combination of substrates produces a convergent electron flow to the ubiquinone pool via CI and CII. To this end, we first added malate, pyruvate and glutamate to generate NADH for the CI. Subsequently, we added the CII substrate, succinate, to obtain CI&CII-linked respiration. By OXPHOS and state 3 we refer to phosphorylating respiration (ATP-producing respiration) in intact mitochondria in presence of saturating ADP. We have revised the main text **(PAGE 9, LINES 209-210)** and figure legends **(Fig. 6I, J)** to clarify the terminology.

3) Normal wt or AOX values for comparison are lacking in some figures; 5g, 6b/c

In healthy wild-type mice, ectopic AOX has been shown to be inert (Szibor et al. 2017), and in line with this we observed very few changes in the "AOX" mouse group compared to WT. This is why we did not include the AOX group in all assays, e.g. the metabolomics. We have added a statement to clarify this in the Methods section **(PAGE 20, LINES 474-475)**.

In the BNGE analyses, the amount of RISP in free CIII₂ and supercomplexes was normal in AOX mice **(Fig 6A)**. Therefore, we do not present the ratio of RISP/CORE2 for the AOX group in **Fig 6B**.

4) Could elevated NO -inhibit cIV activity and thereby OXPHOS in situ?

In our model, the amount of RISP is decreased but it is not completely lost, which allows some electron transfer to cIV. Therefore, increased nitric oxide may theoretically further decrease cIV activity. However, we found no differences between the groups in cIV enzymatic activity in isolated mitochondria (Fig. 6G). This does not rule out an effect *in vivo* (or *in situ*), as the reviewer suggest. Even though our data speak against global ROS or nitrosative damage as an important pathomechanism, they do not rule out a more localized damage, such as to some RC complexes. Future studies should address this question more in detail. We have added a comment on cIV activity in Discussion. **(PAGE 14, LINE 332)**

Discussion -

1) Re tissue specificity; compared wt with GROX- could there be some sort of threshold value that is crucial for liver function? i.e. liver has a priory low activities and is therefore more vulnerable ? What is presently known human/animal models?

This is an important but difficult-to-answer theoretical question. In GRACILE syndrome patients and *Bcs1^{p.S78G}* mice, early histopathology and cIII deficiency are first seen in the liver, but the reason for this is unknown. In the mice, cIII activity is of similar magnitude in liver, kidney and heart (Fig. 6D), but CI&cII-linked phosphorylating respiration an order of magnitude lower in liver than in kidney and heart **(Fig 6I)**, suggesting that the liver has relative low need to maximize ATP production by OXPHOS. This may explain why AOX does not rescue the liver manifestations efficiently. It is also possible that the numerous other anabolic and catabolic special tasks of hepatic mitochondria, including detoxification of numerous compounds, contribute to the pathogenesis in this tissue, but are not affected by the CIII bypass. We have elaborated the Discussion. **(PAGES 12-13, LINES 284-298)**

2) Elaborate on the possibility if /how AOX could be delivered for example by a viral vector to patients?

We have now added discussion on further preclinical studies and on prospects of translatability of our findings to human patients. **(PAGE 15, LINES 340-352)**

Referee #2 (Comments on Novelty/Model System for Author):

This is a very interesting manuscript with relevant pre-clinical data.

Referee #2 (Remarks for Author):

Rajendran et al describes the results of expression of *Ciona intestinalis* alternative oxidase (AOX) in a mouse model of Complex III deficiency. Specifically, this mouse model is a knock-in for the homozygous *Bcs1l*c.232A>G (*Bcs1l*p.S78G) mutation associated with the GRACILE syndrome and recapitulates many of the clinical manifestations, including growth failure, progressive hepatopathy, kidney tubulopathy, and short survival in a C57Bl/6NBomTac background. In this paper the Authors used another genetic background (C57BL/6JCrI) in which the homozygotes for the *Bcs1l*.S78G mutation, survive longer and develop additional later onset phenotypes to test the effects of AOX expression.

The manuscript is well written with interesting data. I have few comments:

-In Figure 1 d-i the histograms show different data obtained both in male and female. However, the body weight is reported only for males. Is there any difference in female?

We have now added weight data of female mice (**Figure 1C**).

-On page 8 this sentence: "AOX had only a minor effect on the hepatic metabolites (Fig. 4b)" refers to figure 4c and not on 4b as indicated.

The error has been corrected.

-The Authors demonstrated that AOX expression rescue pathological phenotype mainly in heart and kidney but not liver. The data reported in figure 6a show that RISP in SC1 and CIII is higher in GRAC vs GROX in kidney and liver, but the enzymatic activity of Complex III is comparable between heart and kidney, but significantly reduced in liver. Is it possible that AOX can't rescue the phenotype of the liver because there is a partially assembled CIII, although not biochemically active? Moreover, western-blot analysis in supplementary figure 4 show that AOX expression is the lowest in liver and kidney. Could this also contribute to the lack of rescue in the liver?

The GRAC mice are, indeed, likely to have a mixture of partially and fully assembled cIII dimers due to poor, but not completely blocked, RISP incorporation in all three tissues studied. A cIII monomer without RISP is unable to oxidize quinols because the first electron transfer (from Q_o quinol to RISP) does not take place. A heterodimer containing one fully assembled monomer (with RISP) may still be active. However, the main trigger for AOX activation is thought to be the high reduction status of the quinone pool (El-Khoury 2014), in our case due to loss of the cIII quinol oxidase activity. Therefore, the assembly status of cIII as such should not affect AOX activity and its consequences. The partial rescue of cIII assembly and activity by AOX we observed in cardiac mitochondria (**Fig. 6A, D**) is interesting but neither our data nor literature offer any clear explanation. It could be linked to the generally improved mitochondrial morphology (**Fig. B-F**), or indirect damage to cIII, (e.g. by ROS or RNS). Conversely, the amount of fully assembled cIII was lower in GROX than in GRAC liver and kidney but the enzyme activity was not significantly changed. In these tissues, the apparent further loss of RISP from cIII in the presence of AOX could simply reflect relaxation of the need to keep cIII assembled and running when AOX replaces its ubiquinol oxidase activity. See Discussion (**PAGES 12-13, LINES 273-275, 284-298**)

Our Western blot results from the plain AOX mice are in agreement with Szibor et al. (2017), indeed showing highest protein expression in heart and lower in kidney and liver. However, AOX protein level was robustly increased by the cIII defect (independently of mitochondrial mass), and this resulted in similar level in all three GROX tissues (**Fig. EV3H**). Therefore, the amount of AOX protein in the affected tissues did not correlate with the rescue effect. We have added a statement to the discussion on this topic. (**PAGE 13, LINES 294-298**).

-I would strongly recommend adding in the discussion the recently published paper by Dogan SA et al "Perturbed Redox Signaling Exacerbates a Mitochondrial Myopathy", Cell metabolism, in view of possible human therapeutic strategy based on AOX expression

As recommended, we have now discussed the very recent findings of Dogan et al. from their cIV deficiency-associated myopathy model. (**PAGE 13-14, LINES 309-325**)

Referee #3 (Remarks for Author):

A very well performed study proposing a natural method to replace defective mitochondrial respiration. The authors propose different damage mechanisms between tissues (energetics vs. ROS damage) to explain different responses to AOX. I would like to see data on transgene expression in different tissues because this could be an alternative explanation.

This is an important point when working with transgenic overexpression. The *AOX* transgene was expressed constitutively under the widely used chicken β -actin promoter (Szibor et al. 2017). We have now added data on AOX mRNA expression, extracted from the transcriptomics data. These data show similar mRNA expression in liver and heart and somewhat lower in kidney. **(Figure EV3H)**

2nd Editorial Decision

26 October 2018

Thank you for the submission of your revised manuscript to EMBO Molecular Medicine. We have now received the enclosed reports from the referees. As you will see the reviewers are now supportive, and I am pleased to inform you that we will be able to accept your manuscript pending following final editorial amendments.

***** Reviewer's comments *****

Referee #1 (Remarks for Author):

The revision is satisfactory and I have no further comments

Referee #2 (Remarks for Author):

The Authors answered to all the questions and the manuscript has been improved.

Referee #3 (Remarks for Author):

Please refer to AOX protein expression in tissues (FIG EV3) in the text. Could the preferential cardiac expression account for the preferential cardiac benefit?

2nd Revision - authors' response

8 November 2018

Referee #3 (Remarks for Author):

Please refer to AOX protein expression in tissues (FIG EV3) in the text. Could the preferential cardiac expression account for the preferential cardiac benefit?

We apologize that we did not clarify this sufficiently in the first revision. AOX mRNA expression was similar in the heart and in the liver (Fig. EV3H), i.e. AOX expression did not correlate at transcriptional level with the rescue. In total tissue lysates from the AOX mice (Fig. EV3H, blue bars), the amount of AOX protein was, indeed, considerably higher in heart than in liver or kidney. However, this difference was mainly due to the much higher mitochondrial mass in heart, as shown by the mitochondrial loading control VDAC1 (Fig. EV3I), and also by the higher amount of most respiratory chain subunits (Fig. EV3I). Independent of this, we observed that the amount of AOX protein was affected by the *Bcs1l* mutation (AOX vs. GROX mice). In GROX liver and kidney, AOX protein was increased whereas in GROX heart it was decreased, resulting in almost identical AOX protein level in these tissues (Fig. EV3I, green bars). In summary, it is very unlikely that the preferential cardiac rescue was due to higher expression of AOX in the heart. We have now added a short paragraph in Results (page 8, lines 168-176) to clarify this.

Corresponding Author Name: Jukka Kallijärvi
Journal Submitted to: EMBO Molecular Medicine
Manuscript Number: EMM-2018-09456-V3